# Robust but weak winter atmospheric circulation response to future Arctic sea ice loss

D. M. Smith [1✉], R. Eade[1], M. B. Andrews[1], H. Ayres [2], A. Clark[1], S. Chripko[3], C. Deser [4], N. J. Dunstone[1], J. García-Serrano[5], G. Gastineau[6], L. S. Graff [7], S. C. Hardiman[1], B. He [8], L. Hermanson [1], T. Jung [9,10], J. Knight[1], X. Levine [11], G. Magnusdottir [12], E. Manzini[13], D. Matei[13], M. Mori [14], R. Msadek[3], P. Ortega [11], Y. Peings [12], A. A. Scaife [1,15], J. A. Screen [15], M. Seabrook[1], T. Semmler [9], M. Sigmond [16], J. Streffing[9,18], L. Sun[17] & A. Walsh[15]

The possibility that Arctic sea ice loss weakens mid-latitude westerlies, promoting more severe cold winters, has sparked more than a decade of scientific debate, with apparent support from observations but inconclusive modelling evidence. Here we show that sixteen models contributing to the Polar Amplification Model Intercomparison Project simulate a weakening of mid-latitude westerlies in response to projected Arctic sea ice loss. We develop an emergent constraint based on eddy feedback, which is 1.2 to 3 times too weak in the models, suggesting that the real-world weakening lies towards the higher end of the model simulations. Still, the modelled response to Arctic sea ice loss is weak: the North Atlantic Oscillation response is similar in magnitude and offsets the projected response to increased greenhouse gases, but would only account for around 10% of variations in individual years. We further find that relationships between Arctic sea ice and atmospheric circulation have weakened recently in observations and are no longer inconsistent with those in models.

[1] Met Office Hadley Centre, Exeter, UK. [2] Department of Meteorology, University of Reading, Reading, UK. [3] CECI, Université de Toulouse, CNRS, CERFACS, Toulouse, France. [4] National Center for Atmospheric Research, Boulder, CO, USA. [5] Group of Meteorology, Universitat de Barcelona, Barcelona, Spain. [6] UMR LOCEAN, Sorbonne Université/CNRS/IRD/MNHN, Institut Pierre Simon Laplace (IPSL), Paris, France. [7] Norwegian Meteorological Institute, Oslo, Norway. [8] State Key Laboratory of Numerical Modeling for Atmospheric Sciences and Geophysical Fluid Dynamics, Institute of Atmospheric Physics, Chinese Academy of Sciences, Beijing, China. [9] Alfred Wegener Institute, Helmholtz Centre for Polar and Marine Research, Bremerhaven, Germany. [10] Institute of Environmental Physics, University of Bremen, Bremen, Germany. [11] Barcelona Supercomputing Center, Barcelona, Spain. [12] Department of Earth System Science, University of California Irvine, Irvine, CA, USA. [13] Max-Planck-Institut für Meteorologie, Hamburg, Germany. [14] Research Institute for Applied Mechanics, Kyushu University, Fukuoka, Japan. [15] College of Engineering, Mathematics and Physical Sciences, Exeter University, Exeter, UK. [16] Canadian Centre for Climate Modelling and Analysis, Environment and Climate Change Canada, Victoria, BC, Canada. [17] Department of atmospheric science, Colorado State University, Fort Collins, CO, USA. [18] Present address: Jacobs University Bremen, Campus Ring 1, 28759 Bremen, Germany. ✉email: doug.smith@metoffice.gov.uk

Since the 1990s the Arctic has been warming more than twice as fast as the global average[1], accompanied by rapid loss of sea ice[2]. This is consistent with polar amplification of climate change and is expected to continue in response to anthropogenic emissions of greenhouse gases[3]. Over the same period, winter temperatures over mid-latitude northern continents, especially Eurasia, have unexpectedly remained steady or cooled, with an apparent increase in severe winter weather[4–7]. The possibility that Arctic warming promotes more severe mid-latitude winters, by altering the atmospheric circulation, has been the subject of intense scientific debate[4,5,8–19]. Observational studies have suggested a clear link between Arctic sea ice loss and mid-latitude winter severity[5,6,20–26], but dedicated numerical model experiments, which are essential to establish causality and to understand the physical mechanisms, are inconclusive, with some simulating mid-latitude cooling in response to Arctic sea ice loss[27–34] and others not supporting this link[16,18,35–40].

For Arctic warming to promote cooling over mid-latitudes would require changes in atmospheric circulation involving a weakening of mid-latitude westerly winds[38], consistent with a negative phase of the North Atlantic Oscillation (NAO), and/or a strengthening of the Siberian High[41,42]. Hence, understanding and quantifying the mid-latitude atmospheric circulation response to Arctic sea ice loss is critical, but there is currently little consensus in modelling studies: the full spectrum of NAO responses has been reported including negative NAO[17,27,29,30,32,41,43,44], positive NAO[45–50], little response[19,51–54] and a response that depends on the details of the forcing[28,31,36,55–58] or the background state of the climate system[59–61].

There are many potential reasons why previous modelling results are inconsistent, including the use of different magnitudes and patterns of imposed sea ice changes, treatment of oceanic feedbacks[41], different models, and whether the simulated responses can be distinguished from internal variability. To overcome some of these limitations, the Polar Amplification Model Intercomparison Project[62] (PAMIP) contribution to the sixth Coupled Model Intercomparison Project[63] (CMIP6) proposed a set of coordinated experiments. Here, we analyse a large ensemble of PAMIP experiments consisting of more than 3000 simulations from 16 different models and find that all models simulate a weakening of mid-latitude tropospheric westerly winds in response to projected Arctic sea ice loss. We elucidate the main physical processes and show that the model spread depends on eddy feedback. This is 1.2 to 3 times too weak in the models, suggesting that the real-world weakening of westerly winds lies towards the higher end of the model simulations. We also show that observed relationships between Arctic sea ice and atmospheric circulation have weakened recently and are no longer inconsistent with those in models. However, the modelled response to Arctic sea ice loss is weak relative to inter-annual variability, though it is similar in magnitude and offsets the projected response to increased greenhouse gases.

## Results

**Multi-model response.** The atmospheric response to future Arctic sea ice loss is diagnosed from two sets of global atmospheric model simulations (Methods). The first set (present-day) is driven by sea surface temperatures (SSTs) and sea ice concentrations (SICs) representing the present-day climate. The second set (future-Arctic) is the same except that Arctic sea ice and coincident SSTs are replaced with values expected if global temperatures rise by 2 °C. By construction, the difference (future-Arctic minus present-day) provides the model-simulated response to future Arctic sea ice loss. We assess 16 model simulations each with between 98 and 300 ensemble members (Table 1) and forced with the same SSTs and SICs. We focus on

the boreal winter season (December, January and February, DJF) for which the imposed sea ice changes show reductions around the edges of the ice pack, especially in the Barents-Kara Seas, Sea of Okhotsk, Bering-Chukchi Seas, and Hudson Bay and Labrador Sea (Fig. 1a). In winter, sea ice insulates the atmosphere from the warmer ocean. Hence warm SSTs are imposed where sea ice is lost in future, producing local maxima of near surface warming response in these regions along with further warming spread throughout the Arctic and into lower latitudes (Fig. 1b).

In the multi-model mean, there are minima in mean sea level pressure (MSLP) response situated over the regions of largest sea ice loss (Fig. 1c), consistent with a thermodynamic heat low response to surface warming. However, there is also a ridge of high pressure extending from Greenland to Siberia with low pressure further south, producing a response that projects onto a negative NAO and a strengthened Siberian High. Although these features are statistically significant in the multi-model ensemble mean (stippling in Fig. 1c), there is disagreement between individual models on the sign of the response in many regions (grey stars show where 90% of models agree).

Consistencies and differences among the models in the dynamical response are further illustrated in the zonally averaged zonal wind ($\bar{u}$, where the overbar denotes the zonal average) response as a function of latitude and height (Fig. 2). In the troposphere, there is a very robust equatorward shift, with a weakening of zonal winds around 55–65°N and a strengthening around 30–40°N simulated by all models. However, the response is much less coherent in the stratosphere, with some models simulating a significant weakening but the majority showing an insignificant response of either sign. This suggests that a stratospheric pathway highlighted in some studies[5,30,33,64–66] is not essential for the sign of the tropospheric and surface response. However, it could act to modulate the magnitude of the surface response, as discussed later. Even in the troposphere where the sign of the response is robust, regions of statistical significance (stippled in Fig. 2) are not consistent across models and the strength of the response varies greatly between models. This raises the key question of what the real-world response would be, and whether the differences between models can be understood in order to derive a constrained estimate. Further progress therefore requires understanding the physical processes.

**Physical processes.** We focus on explaining the robust multi-model average zonal-mean temperature and zonal wind responses in the troposphere. The largest zonal-mean warming ($\bar{T}$) occurs close to the surface over the Arctic (Fig. 3a), consistent with direct heating of the atmosphere by the imposed local surface warming associated with the loss of sea ice. There is a second warming maximum over the Arctic in the lower stratosphere (centred around 100–200 hPa) which has previously been suggested to be heated directly by energy transported from below[5]. However, in the model simulations there is a meridional overturning circulation response with air descending at high latitudes in the lower stratosphere (arrows in Fig. 3a) indicating that this region warms adiabatically as part of the dynamical response[67]. Furthermore, warming in this region is not robust across the models. Hence, the robust weakening of the mid-latitude tropospheric winds is unlikely to be solely caused by a simple reduction of meridional temperature gradient by heating of the high latitude atmosphere from below.

A meridional overturning circulation response is also seen in the mid-latitude lower troposphere (Fig. 3a) and, as we show below, is important for understanding the physical mechanisms and explaining the spread in modelled responses. This circulation is thermally indirect, with air rising over relatively cool surface

**Table 1 Models, ensemble sizes and data used.**

| Institute | Model | Resolution[a] | Ensemble size | Data | EP flux frequency[b] |
|---|---|---|---|---|---|
| National Center for Atmospheric Research | CESM2 | 1.25 × 1.0 × 32 × 2.25 | 200 | All (only 100 members for EP fluxes) | 30 min |
| Canadian Centre for Climate Modelling and Analysis | CanESM5 | 2.8 × 2.8 × 49 × 1 | 300 | All (only 100 members for tas and psl) | 6-hourly |
| Centre Européen de Recherche et de Formation Avancée en Calcul Scientifique | CNRM-CM6-1 | 1.0 × 1.0 × 91 × 0.01 | 300 | All | daily |
| US Department of Energy/University of California Irvine | E3SMv1 | 1.0 × 1.0 × 72 × 0.12 | 200 | All except $F_\phi$, $F_P$ for experiment 1.6 and $\bar{v}^*$, $\bar{w}^*$ for both | daily |
| Barcelona Super Computing Centre | EC-EARTH3 | 1.0 × 1.0 × 91 × 0.01 | 150 | All | 6-hourly |
| Alfred Wegener Institute Bremerhaven; Max Planck Institute for Meteorology, Hamburg | ECHAM6.3 | 0.94 × 0.94 × 95 × 0.01 | 100 | All | 6-hourly |
| Institute of Atmospheric Physics, Beijing | FGOALS-f3-L | 1.0 × 1.0 × 32 × 2.19 | 100 | All except $F_P$ | daily |
| Met Office UK | HadGEM3-GC31-MM | 0.55 × 0.83 × 85 × 0.005 | 300 | All | 20 min |
| University of Exeter | HadGEM3-GC31-LL | 1.25 × 1.875 × 85 × 0.005 | 195 | All | 20 min |
| Institute Pierre Simon Laplace | IPSL-CM6A-LR | 1.26 × 1.25 × 79 × 0.01 | 200 | All | daily |
| University of Tokyo/National Institute for Environmental Studies/Japan Agency for Marine-Earth Science and Technology | MIROC6 | 1.4 × 1.4 × 81 × 0.004 | 100 | All | daily |
| Norwegian Meteorological Institute | NorESM2-LM | 1.9 × 2.5 × 32 × 3.6 | 200 | All | 30 min |
| Alfred Wegener Institute | OpenIFS-159 | 1.125 × 1.125 × 91 × 0.01 | 299 | All | 6-hourly |
| Alfred Wegener Institute | OpenIFS-511 | 0.352 × 0.352 × 91 × 0.01 | 98 | All | 6-hourly |
| Alfred Wegener Institute | OpenIFS-1279 | 0.14 × 0.14 × 137 × 0.01 | 100 | All except $\bar{v}^*$, $\bar{w}^*$, $F_P$ | 6-hourly |
| National Center of Atmospheric Research/University of California Irvine | CESM1-WACCM-SC | 1.9 × 2.5 × 66 × 5.9 × 10$^{-6}$ | 300 | All except $\bar{v}^*$, $\bar{w}^*$ | daily |

a (degrees latitude) × (degrees longitude) × (number of vertical levels) × (lid height, hPa)
b The sampling frequency for the calculation of EP fluxes

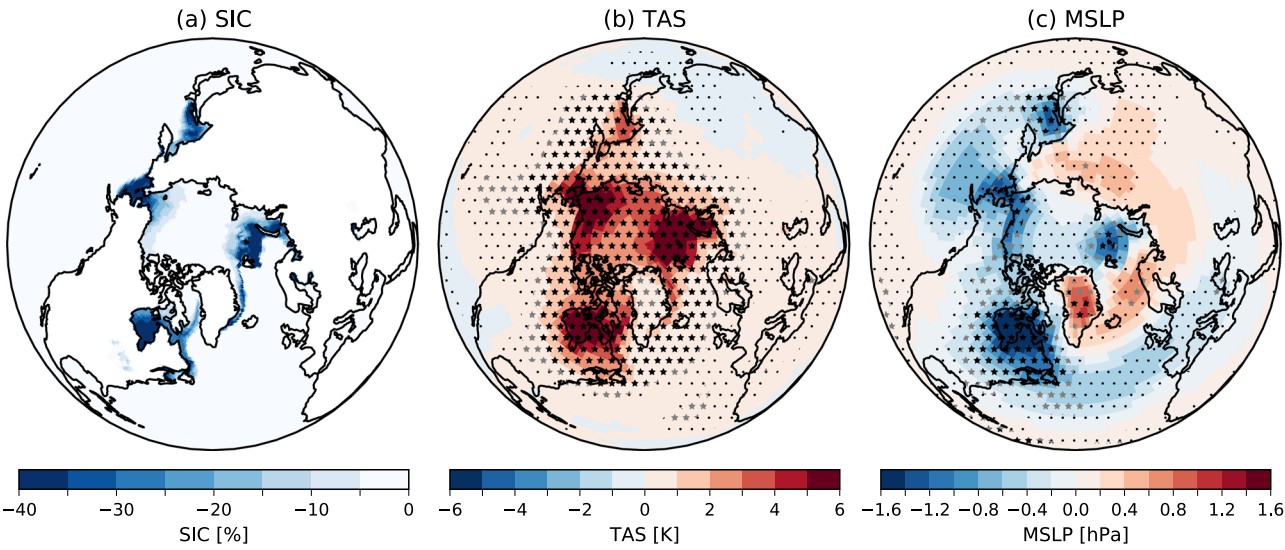

**Fig. 1 Winter response to future Arctic sea ice loss. a** Imposed winter sea ice concentration difference (%). **b** Near surface temperature (TAS) response (K). Note that surface temperature changes are imposed in regions of sea ice loss. **c** Mean sea level pressure (MSLP) response (hPa). All plots show the winter (December, January, February, DJF) mean, and responses are for the multi-model ensemble mean (calculated as the unweighted average of all ensemble members). Stippling indicates where the multi-model ensemble mean response is significant (95% confidence interval). Black (grey) stars indicate where 100% (90%) of the individual models agree on the sign of the response.

conditions between 35–50°N and descending over the warmer surface between 55 and 70°N. It is therefore not a direct thermally driven response to the imposed high latitude warming and, as we show below, highlights a key role for changes in wave (eddy) activity.

Climatologically, transient wave activity is mostly generated near the surface by baroclinic eddies in the storm tracks around 40–60°N and propagates vertically and meridionally until the waves break and dissipate[68]. Waves flux angular momentum in the opposite direction to their propagation, accelerating zonal winds in their source regions and decelerating zonal winds in their dissipation regions. Wave propagation may be depicted by Eliassen-Palm (EP) fluxes (Methods equation 6), with regions of EP flux divergence indicating where waves act to accelerate the zonal flow and regions of EP flux convergence indicating where waves act to decelerate the zonal flow (Methods equations 2 and 5).

The multi-model mean wave activity response (Fig. 3c) mainly consists of an equatorward shift of upward EP flux ($F_p$) in the troposphere, with increased $F_p$ around 35–55°N and decreased $F_p$ around 60–75°N, and an increased northward EP flux ($F_\phi$) in the mid to upper troposphere connecting these latitudes. The resulting divergence of $F_\phi$ accelerates the zonal wind around 30–40°N and convergence of $F_\phi$ decelerates the zonal wind around 55–65°N (Fig. 3d), consistent with the robust equatorward shift of the winds noted above. We will show below that the strength of the EP flux response is important for explaining the model spread, and focus first on understanding its origins.

Several previous studies[30–32,36,69–71] have highlighted an increase in $F_p$ in response to Arctic sea ice loss. However, the cause is unclear given that the reduced meridional temperature gradient between the equator and the pole (Fig. 3a) would weaken the baroclinic generation of eddies in the storm tracks[72,73] and hence reduce $F_p$, as seen in other studies[43,61,67,74]. There is a small increase in $F_p$ north of 80°N (Fig. 3c) that is consistent with zonal asymmetries in near surface temperature and sea level pressure response over regions of sea ice loss (Fig. 1), but the main signal is an equatorward shift resulting in an increase in $F_p$ around 35–55°N that also extends into the stratosphere. Hence, understanding the main physical processes involved in this equatorward shift is key for understanding the mid-latitude atmospheric response to Arctic sea ice loss.

Although causality cannot be unequivocally determined, much insight can be gained by considering the evolution from autumn to winter, focussing on processes that are robustly simulated across the models. In October there is a weakening of both $\bar{u}$ and $F_p$ around 50–75°N (Fig. 4b, c) consistent with the imposed reduction in meridional surface temperature gradient, whereas an equatorward shift is first seen in November (Fig. 4f, g) and develops into the DJF pattern (Fig. 3). To explain this evolution we highlight the following processes which appear to be simulated by the majority of models, though we note that they do not explain every aspect of the circulation response and other processes[75] are also likely operating:

1. Reduced zonal wind shear and eddy formation. Arctic warming decreases the surface meridional temperature gradient in October (Fig. 4a), reducing the wind shear on the poleward side of the jet (around 60–70°N, Fig. 4b) via the thermal wind relation (Methods equation 1). Reduced wind shear reduces baroclinic eddy formation, weakening the storm track and reducing $F_p$ at the surface around 50–75°N (Fig. 4c).

2. Meridional overturning circulation. A lower tropospheric mid-latitude meridional overturning circulation anomaly (Fig. 4a, e, highlighted above) develops between October and December such that the resulting flow maintains thermal wind balance and is consistent with changes in eddy activity. Some aspects of this circulation can be understood by considering that the reduced $F_p$ at the surface around 50–75°N (Fig. 4c, g) results in a positive $\nabla_p F_p$ immediately above (Fig. 4d, h) because the flux into this region reduces more than the flux out of this region. An increase in $\nabla_p F_p$ must be balanced (Methods equation 2) by an increase in zonal wind and/or an equatorward flow (negative $\bar{v}^*$). However, zonal wind tends to be reduced in response to the imposed weakening of the surface temperature gradient (Fig. 4b) and $\nabla_p F_p$ is at least partly balanced by an equatorward flow near the surface around 45–55°N (Fig. 4a, e). To maintain mass continuity, a meridional circulation develops with ascent further equatorward (35–50°N), poleward flow in the mid to upper troposphere (50–70°N) and descent around 65–75°N.

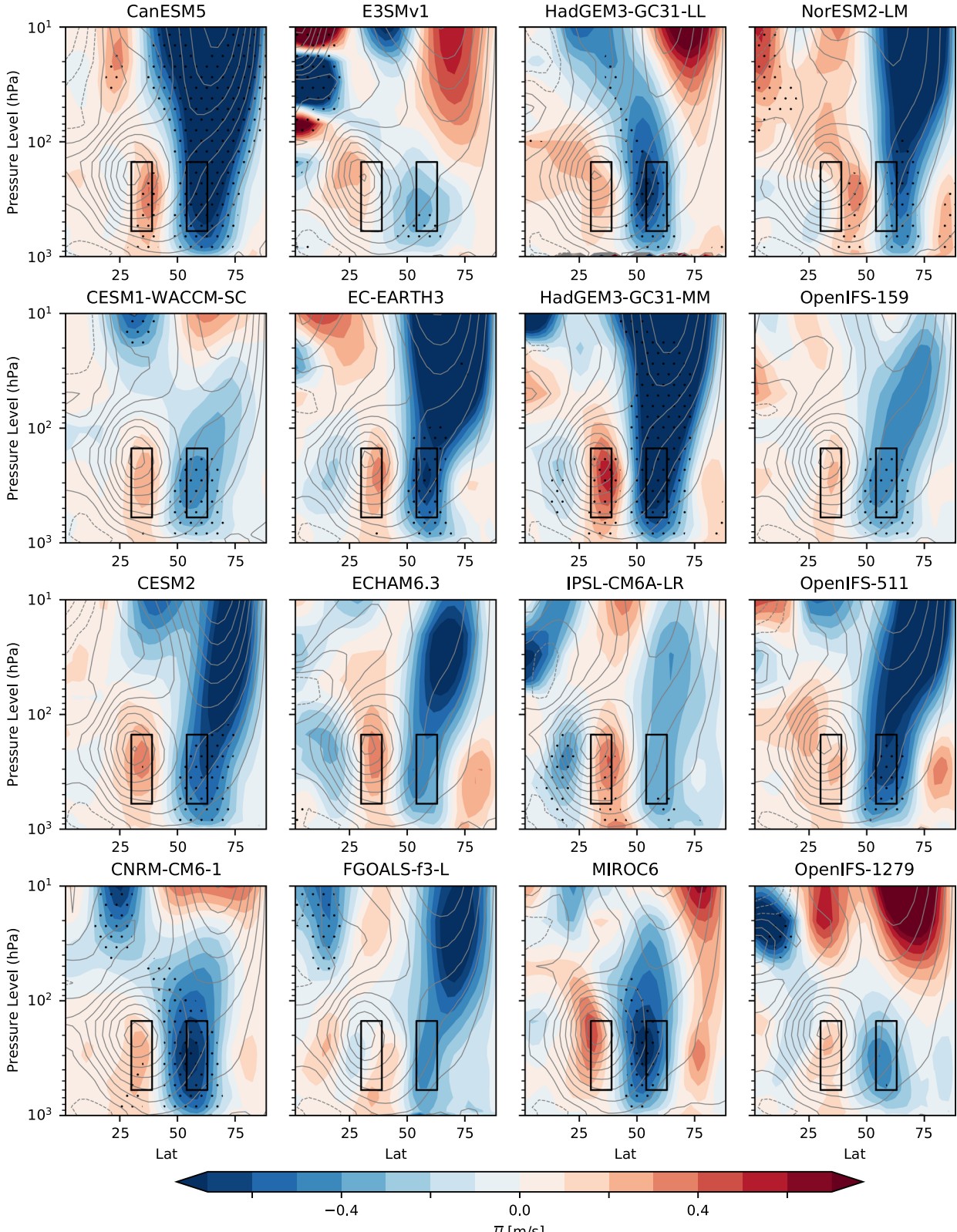

**Fig. 2 Consistent tropospheric response to Arctic sea ice loss.** Zonally averaged DJF zonal wind response ($\bar{u}$, $ms^{-1}$, where the overbar denotes the zonal average) plotted as a function of latitude (°N) and height (pressure) for the ensemble mean of each of the models. The boxes show the regions used to compute the zonal wind response index (ZWRI). Stippling indicates where the ensemble mean response is significant (95% confidence interval). Contours show the climatological zonal mean winds (contour interval 5 $ms^{-1}$ with negative contours dotted).

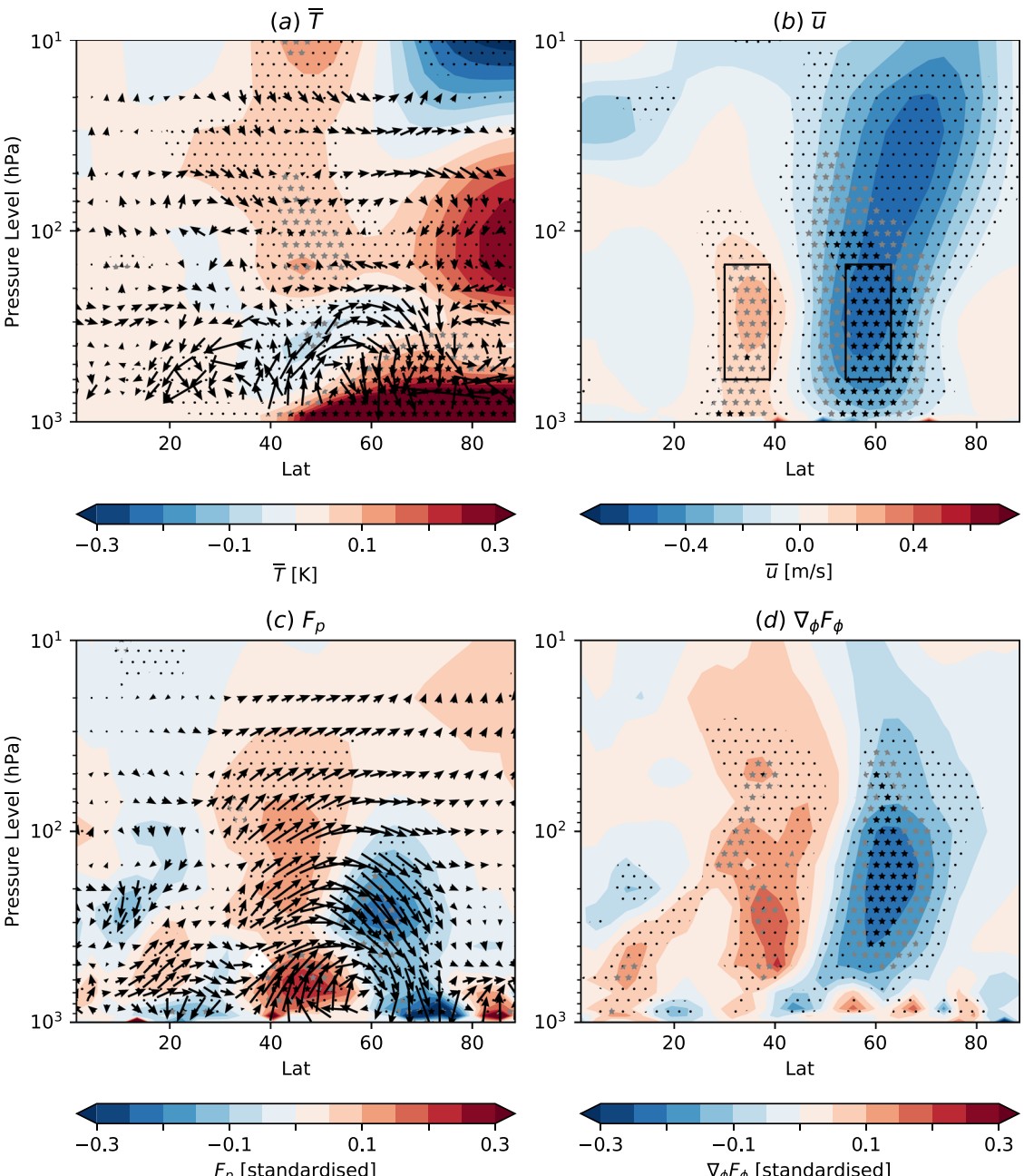

**Fig. 3 Zonal mean response.** Latitude-height cross sections of the multi-model ensemble mean zonally averaged DJF response of **a** temperature ($\bar{T}$, colours, K, where the overbar denotes the zonal average) and transformed Eulerian mean (TEM, Methods) circulation (arrows), **b** zonal wind ($\bar{u}$, $ms^{-1}$), and **c** upward Eliassen-Palm (EP) flux ($F_p$, colours, standard deviations) and EP flux vectors (arrows representing $F_\phi$ and $F_p$), **d** divergence of northward EP flux ($\nabla_\phi F_\phi$, standard deviations). Stippling as in Fig. 1. To aid visualisation the TEM circulation and EP fluxes are standardised by dividing by the internal variability of the present-day simulations (Methods).

Adiabatic cooling of the ascending air around 35–50°N would tend to enhance the meridional temperature gradient and, via thermal wind, strengthen the wind shear further to the south, and reduce the meridional temperature gradient and weaken the wind shear further to the north. Hence, this meridional circulation tends to promote an equatorward shift of the storm track and source of $F_p$, further reducing $F_p$ on the poleward side of the jet and producing an increase in $F_p$ on the equatorward side as seen in the DJF pattern of wave activity response (Fig. 3c).

3. Positive eddy feedback. The wave activity response (Fig. 3c) results in a divergence of $F_\phi$ on the equatorward side of the

jet and convergence of $F_\phi$ on the poleward side of the jet (Fig. 3d) that reinforces the equatorward shift of the storm track and hence enhances the response. Increased $F_p$ can also propagate into the stratosphere, weakening the polar vortex and subsequently further enhancing the equatorward shift by weakening the tropospheric winds on the poleward side of the jet, especially in late winter.

**Sensitivity across models.** We further assess the processes described above by investigating whether they explain the sensitivity of the response across the models. To quantify the response

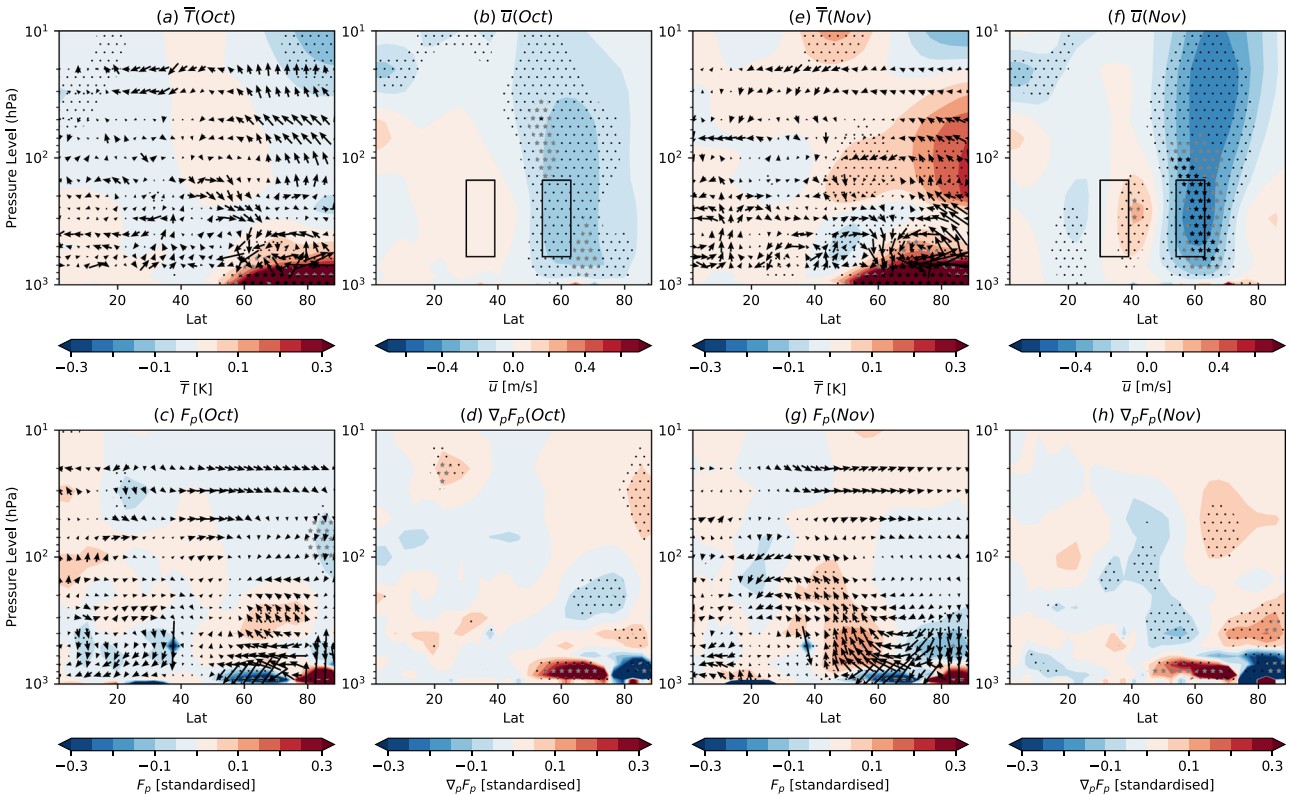

**Fig. 4 Physical mechanisms of zonal mean response.** Latitude-height cross sections of the multi-model ensemble mean zonally averaged October response of **a** temperature ($\overline{T}$, colours, K) and TEM circulation (arrows), **b** zonal wind ($\overline{u}$, $ms^{-1}$), **c** upward EP flux ($F_p$, colours) and EP flux vectors (arrows representing $F_\phi$ and $F_p$), **d** divergence of upward EP flux ($\nabla_p F_p$). (e,f,g,h) As (**a**, **b**, **c**, **d**) but for November. Stippling as in Fig. 1. To aid visualisation the TEM circulation and EP fluxes are standardised by dividing by the internal variability of the present-day simulations (Methods).

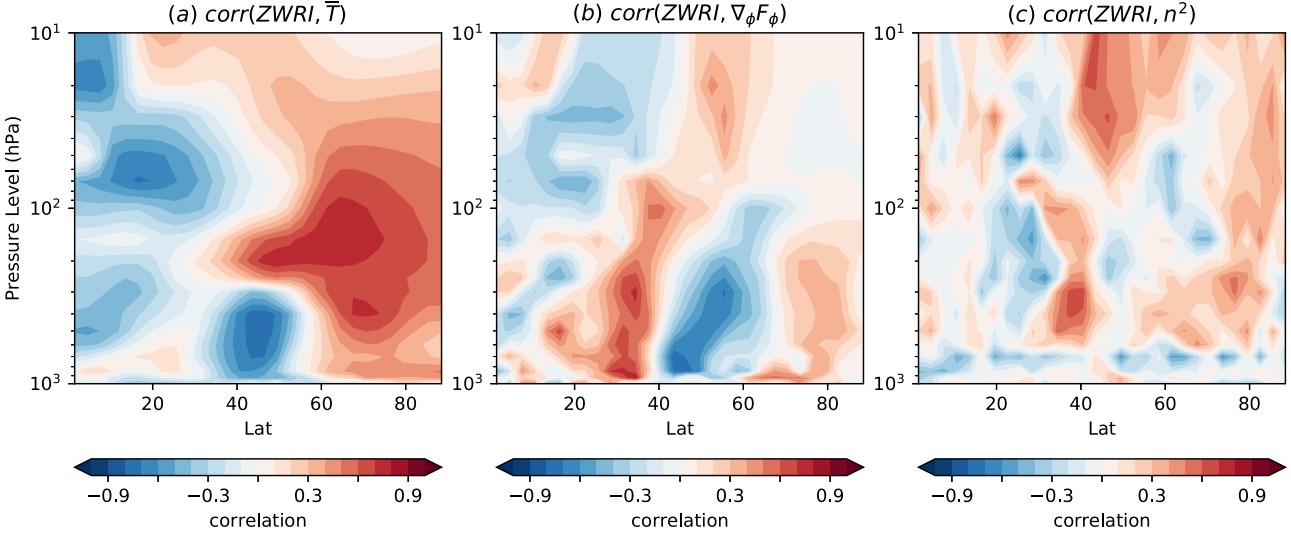

**Fig. 5 Sensitivity of zonal mean response.** Latitude-height cross sections of the correlation across models between ZWRI and response in (**a**) $\overline{T}$, **b** $\nabla_\phi F_\phi$ and **c** refractive index ($n^2$). All data are for DJF.

for each model, we define a zonal wind response index (ZWRI, Methods, the difference between the zonally averaged zonal wind responses in the boxes in Figs. 2 and 3b). The correlation between ZWRI and zonal mean temperature response across the models (Fig. 5a) is consistent with a sensitivity to the strength of the meridional circulation (process 2, Fig. 3a): models with a larger ZWRI show enhanced lower tropospheric adiabatic cooling by ascent around 35–50°N and warming by descent around 65–75°N compared to models with a smaller ZWRI. In addition, models

with a larger ZWRI show greater mid to upper tropospheric warming due to enhanced southerly advection around 50–70°N compared to models with a smaller ZWRI.

ZWRI is also correlated with the divergence of $F_\phi$ around 30–40°N and convergence of $F_\phi$ around 45–65°N (Fig. 5b) consistent with a positive feedback from wave driving in determining the magnitude of the simulated response (process 3). The tropospheric response may also be enhanced by eddy feedback involving the stratosphere (process 3): increased $F_p$ enters the stratosphere around 35–55°N

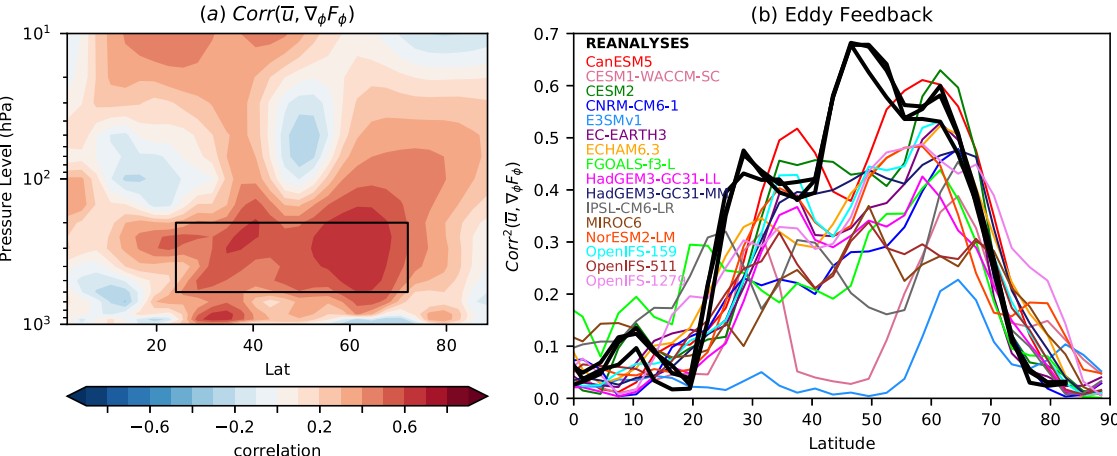

**Fig. 6 Eddy feedback. a** Latitude-height cross section of the multi-model mean local correlation between DJF $\bar{u}$ and $\nabla_\phi F_\phi$. Correlations are computed across ensemble members for the present day simulation for each model separately, and then averaged to make the multi-model mean. **b** Fraction of mid-upper troposphere zonal wind variance explained by $\nabla_\phi F_\phi$ as a function of latitude (the square of the local correlations in **a** averaged over 600 to 200 hPa) for each of the models (coloured curves) and in the reanalyses (black curves). Reanalyses values are computed from interannual time series over the period 1979–2016 inclusive. The eddy feedback parameter (M) is computed as the average over latitudes 25–72°N (shown by the box in **a**).

(Fig. 3c) and ZWRI is correlated with increases in the refractive index ($n^2$, Methods equation 7) in these latitudes (Fig. 5c), allowing more waves to weaken the stratosphere and affect the troposphere via downward propagation, strengthening the negative NAO and equatorward jet shift through further eddy feedback[76].

**Constrained response**. The model spread can potentially be exploited to obtain an estimate of the true response using the concept of emergent constraints (Methods). In this approach, the key physical processes that explain the differences between modelled responses must be understood and related to quantities that can be observed. If a robust relationship exists, then the true response may be inferred by comparing observations of these quantities with those in the models. Results above suggest an important role for eddies: the DJF zonal wind response is initiated by thermal wind balance but is sensitive to eddy feedback which alters $\nabla_\phi F_\phi$ in the troposphere (Figs. 3d and 5b) and potentially involves the stratosphere via changes in $F_p$ and $n^2$ (Figs. 3c and 5c). Eddies are generated by the mean flow but can also feedback onto the mean flow in such a way that increases or reduces their generation (i.e. a positive or negative feedback[77,78]). Hence, we hypothesise that the response to sea ice loss may be related to the strength of the eddy feedback simulated by the different models.

We estimate eddy feedback by examining the role of eddies in driving internal variability (Methods). For each model, we compute the local correlation across the ensemble members between DJF zonal mean zonal wind ($\bar{u}$) and the divergence of northward EP flux ($\nabla_\phi F_\phi$) in the present-day simulations. Averaged across the models, this correlation is positive throughout most of the troposphere (Fig. 6a) consistent with acceleration of the mean flow by a convergence of eddy momentum flux. The square of this correlation represents the fraction of the total variability of $\bar{u}$ that is explained by variations in $\nabla_\phi F_\phi$ and it would be expected to increase as the eddy feedback becomes more strongly positive. The squared correlation varies greatly across the models (Fig. 6b), suggesting a wide range of eddy feedbacks simulated by the different models.

We define a measure of eddy momentum feedback strength (M) as the variance of DJF $\bar{u}$ explained by $\nabla_\phi F_\phi$ (i.e. the squared correlation) averaged over the mid to upper extratropical troposphere in the region shown by the box in Fig. 6a (Methods). We find that M is positively correlated with ZWRI across models

($r = 0.49$, $p = 0.03$, Fig. 7a), supporting our hypothesis that the response to sea ice loss is strengthened by eddy feedbacks and allowing a constrained estimate of the zonal wind response to be obtained via the ensemble regression approach (ER, Methods). We diagnose the observed eddy feedback using three reanalyses and computing the correlations in time rather than across ensemble members (Methods). The observed eddy feedback is between 1.2 and 3 times larger than that in any of the climate models. Using the observed eddy feedback strength to scale the zonal wind responses in the models, we find that the multi-model ensemble mean ZWRI increases from 0.7 ($\pm 0.1$) $ms^{-1}$ to 0.9 ($\pm 0.2$) $ms^{-1}$ (95% confidence intervals) (Fig. 7a, green shading), suggesting that the real world response would lie towards the higher end of the model simulations.

We now apply the ER method in a more general sense to assess the response to sea ice loss in different quantities and regions: for any variable and location, we regress our estimate of eddy momentum feedback strength against the response in that variable and use the observed eddy feedback to diagnose the constrained response. Note, that ER will have little impact where the regression is small. We find that ER enhances the zonal wind response throughout the atmosphere, including the stratospheric polar vortex (Fig. 7b, $r = -0.44$, $p = 0.04$) which increases in magnitude from 0.1 to $-0.6$ $ms^{-1}$ in the simple ensemble mean (EM) to $-0.1$ to $-1.7$ $ms^{-1}$ in ER, accompanied by an increase in regions showing a significant zonal wind response (compare Fig. 8c, d). This is consistent with an increased refractive index response in ER particularly around 35–40°N (Fig. 5c) in the upper troposphere, which allows more waves to propagate into the stratosphere and weaken the SPV (Fig. 8a, b). It is well established that changes in the stratosphere propagate downwards and affect the troposphere and surface winds[76], and this stratospheric pathway likely enhances the near surface wind response. Overall, ER results in a greater reduction of the SPV (Fig. 8c, d), stronger equatorward shift of the storm tracks (Fig. 8e, f), and a stronger negative NAO response (increased from $-0.4$ to $-1.2$ hPa in EM to $-0.3$ to $-2.1$ hPa in ER). However, ER does not constrain the Eurasian temperature response (Eurasia T, Methods), which ranges from $-0.2$ to $+0.4$ °C.

We assess the robustness of our proposed constraint to several sources of uncertainty. The simulated response to Arctic sea ice loss is small relative to internal variability, leading to substantial uncertainties in individual models (ellipses in Fig. 7).

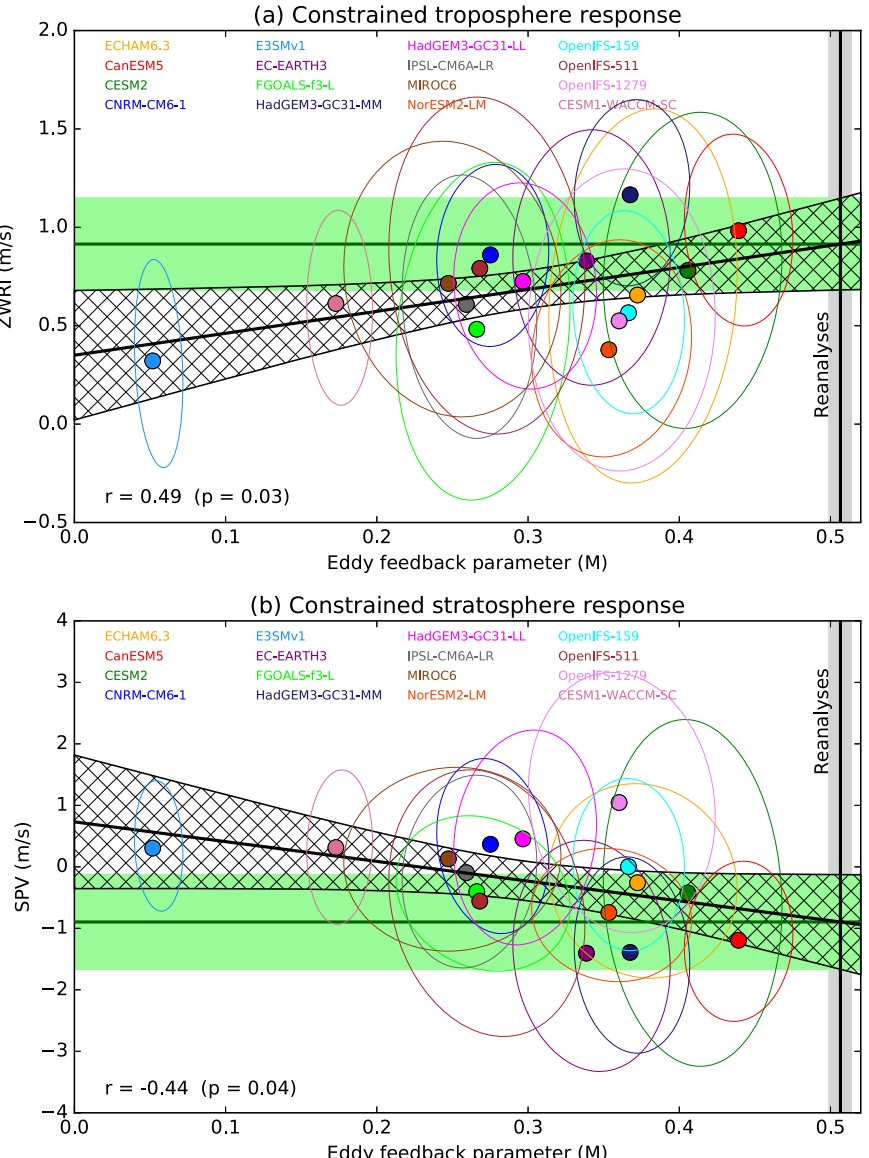

**Fig. 7 Emergent constraints. a** Emergent constraint based on the ensemble regression (ER) between eddy momentum feedback and the zonal wind response index (ZWRI). Black line shows the regression with hatching showing the 95% confidence interval. Horizontal green line shows the constrained ensemble mean response, with the shading showing its 95% confidence interval (Methods). Vertical black line and grey shading shows the mean and range of eddy feedback from the reanalyses. Ellipses show the 95% uncertainties obtained by bootstrapping with replacement the ensemble members. **b** As **a** but for the stratospheric polar vortex (SPV) response. A one-sided test is used to calculate p values since we expect the response to increase as eddy feedback strengthens. All data are for DJF.

We therefore further assess the statistical significance of the regressions by bootstrapping the individual members for each model with replacement. This is repeated 1000 times and results in p values that are slightly less significant ($p = 0.08$ and 0.06 for ZWRI and SPV respectively), highlighting the need for very large ensembles to obtain robust results[79]. We also tested the sensitivity of ER to outliers[80] by removing each model in turn and repeating the regression. This is most sensitive to removing E3SMv1 and CanESM5, which increases the p values to 0.16 and 0.07 respectively for ZWRI, and to 0.06 and 0.10 respectively for SPV. The calculation of EP fluxes can be sensitive to the frequency of data used, which ranges from 20 min to daily means depending on the model (Table 1) and 6 h in the reanalyses[81]. We tested this sensitivity by recalculating the regressions (Fig. 7) for subsets of models with similar sampling frequencies (20–30 min, 6 hourly, and daily) and found the sign to be the same for each

subset as for the full set (though the regressions were no longer significant at the 95% confidence level). The eddy feedback parameters computed from the Reanalyses are based on 38 years compared to more than 100 members for the models. However, the three reanalyses are in good agreement and together provide a similar sample size to the models. Finally, the eddy feedback parameters for reanalyses are based on time series that include the effects of coupled modes of internal variability such as El Niño, in addition to changes in natural and anthropogenic radiative forcings that are not present in the model experiments. We recalculated the eddy feedback using time series from AMIP simulations that also include these factors for five models (CanESM5, HadGEM3-LL, HadGEM3-MM, IPSL-CM6A-LR, MIROC6) and found a small increase (0.09 on average, compared to a multi-model mean of 0.30), but that all model values remain lower than the reanalyses.

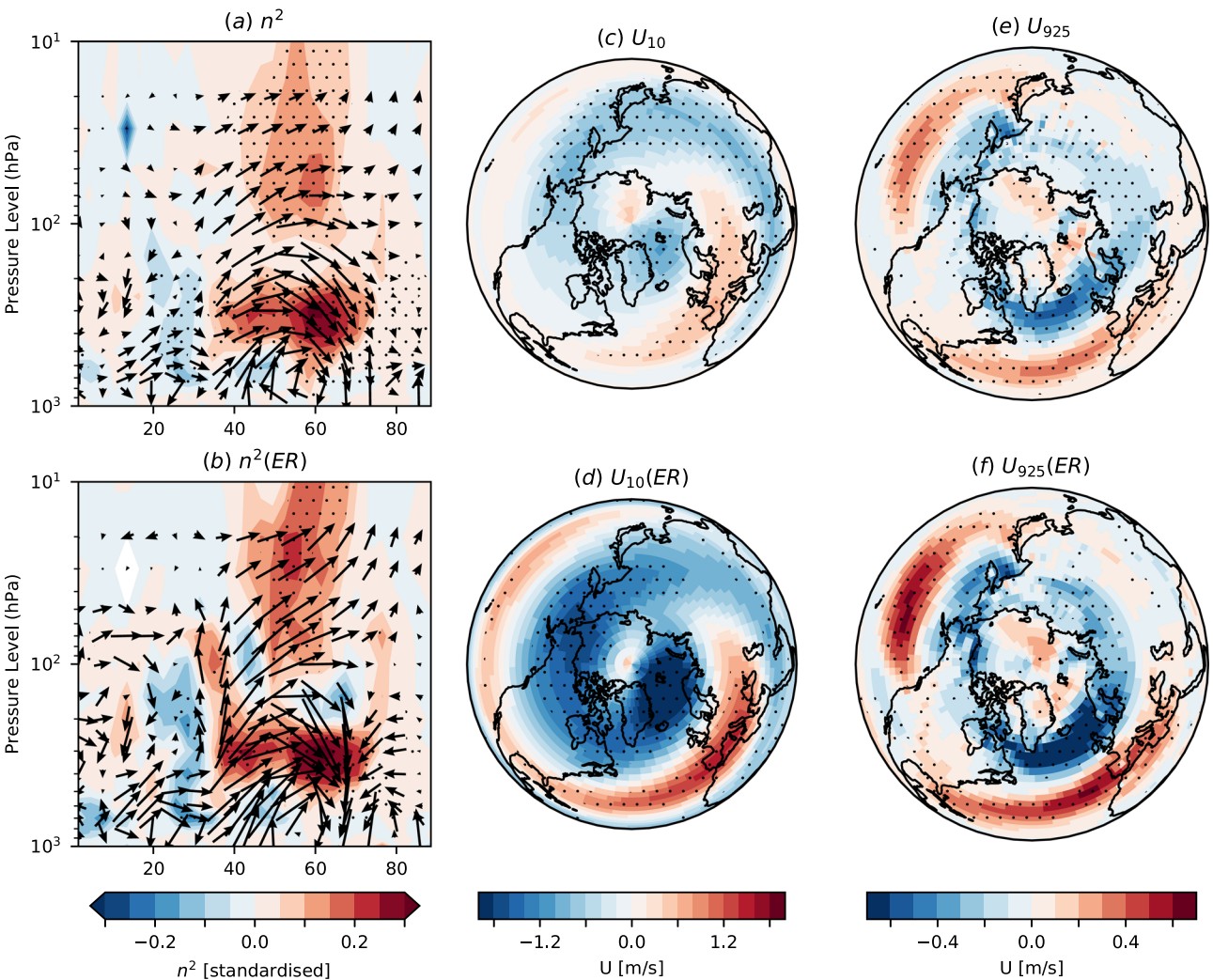

**Fig. 8 Impact of emergent constraint.** Refractive index ($n^2$) response (standardised) for **a** the multi-model ensemble mean (EM) and **b** the ensemble regression (ER). **c**, **d** As **a**, **b** but for stratospheric zonal wind ($ms^{-1}$) at 10 hPa. **e**, **f** As **a**, **b** but for near-surface zonal wind ($ms^{-1}$) at 925 hPa. Arrows in **a** and **b** show the EP flux response (standardised). Colour bars represent each column. Stippling shows where the EM or ER response is significant (95% confidence interval, Methods). All data are for DJF.

Our proposed emergent constraint is therefore reasonably robust and is strongly linked to the physical processes. However, we note that other emergent constraints are possible. For example, ZWRI is correlated with the background SPV ($r = 0.50$, $p = 0.02$, Supplementary Fig. 1) but in this case the observations are near the middle of the model range so that ER is close to the simple ensemble mean. Had we only considered this relationship we would have erroneously constrained the response to be near the EM, highlighting the importance of basing constraints on physical processes[61,82]. It is also possible that the response could be further enhanced by coupling with the ocean[17,41,44], albeit with greater variability[79], and/or by the pattern of sea ice loss[36,57,58,83], and could depend on the phase of the quasi-biennial oscillation[84]. However, initial results from a subset of models for PAMIP experiments that are designed to investigate the effects of coupling and the pattern of sea ice loss (Methods) do not show consistent differences in ZWRI across models compared to the standard experiments. This suggests that these effects may not be large, but results from more models are needed for further assessment, and could provide an important out of sample test[82] of the emergent constraint proposed here.

## Discussion

We have analysed the boreal winter atmospheric circulation response to future Arctic sea ice loss using a very large multi-model ensemble, comprising 16 different models and more than 3000 ensemble members with the same experimental protocol. We find a robust response in the troposphere, with a weakening of mid-latitude tropospheric westerly winds and an equatorward shift of the storm tracks, consistent with a negative phase of the NAO. However, the strength of the modelled response varies between models and is proportional to the strength of the eddy momentum feedback, enabling a constrained estimate of the real-world response to be obtained. Since all the models underestimate the observed eddy feedback strength, the real-world response is likely to be at the higher end of the model range. The stratosphere response is not consistent across the models and therefore not essential for determining the sign of the tropospheric circulation response to sea ice loss. However, the eddy feedback constraint indicates a robust weakening of the stratospheric polar vortex, suggesting that in the real world the stratosphere may play an active role that amplifies the surface response, consistent with other studies[30,33,36,64,66,74].

Our emergent constraint suggests that models may underestimate the response to sea ice loss and is consistent with other evidence that models underestimate the predictable fraction of NAO variability in seasonal[85–87], interannual[88], and decadal forecasts[89–91], and in historical climate simulations[92–94]. This has been referred to as the signal-to-noise paradox[95] since models are unexpectedly able to predict the real world better than they can predict one of their own ensemble members. We find that the eddy momentum feedback is underestimated in all of the models, suggesting that this could be a potential cause of this model error as previously suggested[96]. Possible reasons for underestimated eddy feedback include errors in wave propagation, imperfect representation of interactions with the mean flow, and unresolved waves, but further analysis is left for future work. We also note that our emergent constraint could potentially explain some of the model spread in jet shifts forced by a variety of factors, including changes in greenhouse gases, but this is beyond the scope of the present study.

Diagnosing the response to sea ice loss is not possible from observations alone since causality cannot be established[61]. Nevertheless, there is a perception that models and observations do not agree[5]. Our results suggest that model errors could be responsible for some of the apparent inconsistencies and that taking these into account yields a robust modelled response in the troposphere and stratosphere. However, it is important to note that the short observational record contains considerable sampling uncertainty[97,98]. For example, the period 1979 to 2012 suggests a strong link between autumn sea ice and winter atmospheric circulation and Eurasian temperature (Fig. 9a, c, e) consistent with previous studies[6,20–25]. If we extend the analysis period by adding the most recent 8 years of data (2013–2020), we find that the observed relationships between autumn sea ice extent and DJF NAO, SPV and Eurasian T indices are weaker in magnitude (Fig. 9b, d, f). This is consistent with recent evidence that the observed relationships are modest[7] and intermittent[66] and may not reflect a causal link with Arctic sea ice[99–101].

Our model estimates are within, and therefore consistent with, the magnitudes of the observed relationships based on the extended period of record (1979–2020). However, the modelled response to sea ice loss remains relatively weak, amounting to 30% or less of the observed interannual standard deviation ($\sigma$) of the NAO and SPV. Assuming linearity and given that the imposed reduction in DJF Arctic sea ice extent in our experiments is around $4\sigma$, this implies that interannual Arctic sea ice variations account for less than 10% of the interannual variations in NAO and SPV, and are thus unlikely to drive large seasonal mean impacts in individual winters. Nevertheless, these values are similar in magnitude to changes in the NAO and SPV expected by the end of the century in response to increases in greenhouse gases[65,102,103], and will therefore impact long term projections.

## Methods

**Model experiments**. We assess coordinated experiments from the Polar Amplification Model Intercomparison Project[62] (PAMIP). PAMIP experiment 1.1 simulates the present day climate using global atmosphere models constrained at the surface by present day estimates of sea surface temperature (SST) and sea ice concentration (SIC). PAMIP experiment 1.6 is the same as 1.1 except that Arctic SIC is replaced with values expected with global temperatures 2 degrees Celsius warmer than pre-industrial conditions. Where sea ice is lost in 1.6 relative to 1.1, future SSTs are used; elsewhere the SSTs are the same in 1.6 and 1.1. By construction, the difference between 1.6 and 1.1 provides the simulated response to future Arctic sea ice loss and associated local changes in SST. Each experiment starts on 1st April 2000 and runs for 14 months. We analyse results from 16 models, each with at least 98 ensemble members (Table 1) and forced with the same SSTs and SICs. We also analyse a small subset of models to assess the effects of ocean-atmosphere coupling (PAMIP experiments 2.1 and 2.3, EC-EARTH3, HadGEM3-MM, NorESM2-LM) and regional sea ice changes (PAMIP experiments 3.1 and 3.2, CNRM-CM6-1, ECHAM6.3, EC-EARTH3, HadGEM3-MM). All data were re-gridded to the resolution of the coarsest model (3° latitude by 3° longitude) before comparison.

**Observations**. We use Eliassen-Palm (EP) fluxes from three reanalyses (ERA-Interim, NCEP-NCAR and JRA-55) available from the Centre for Environmental Data Analysis[81]. Sea ice observations are taken from HadISST1.1[104]. Sea level pressure, near-surface temperature and stratospheric wind observations are taken from the ERA5 reanalysis[105].

**Indices**. The North Atlantic Oscillation (NAO) index is calculated as the difference in mean sea level pressure between two small boxes located around the Azores (28–20°W, 36–40°N) and Iceland (25–16°W, 63–70°N) with the average over the whole time series removed to create anomalies[88]. To quantify the response to Arctic sea ice loss, we define a Zonal Wind Response Index (ZWRI) that is calculated as the difference in zonally averaged zonal wind response between 30–39°N and 54–63°N averaged over 600 to 150 hPa, which correspond to the regions with the largest zonal wind responses in the multi-model mean. The strength of the stratospheric polar vortex (SPV) is computed as the zonal-mean zonal wind averaged over 54–66°N at 10 hPa. Eurasian temperatures (Eurasia T) are averaged over the region[34] 60–120°E, 40–60°N. The Barents–Kara (BK) Seas region[58] is taken as 10–100°E, 65–85°N.

**Thermal wind**. The geostrophic balance between pressure gradient and Coriolis forces, combined with the hydrostatic equation leads to the thermal wind relationship, showing that a reduction in meridional temperature gradient as the Arctic warms is accompanied by a decrease in vertical wind shear[68]:

$$f\frac{\partial u}{\partial p} = \frac{R}{ap}\frac{\partial T}{\partial \phi} \qquad (1)$$

where $u$, $T$, $p$, $\phi$ are zonal wind, temperature, pressure and latitude, $f$ is the Coriolis parameter, $a$ is the radius of the Earth and $R$ is the gas constant.

**Transformed Eulerian Mean momentum equation and Eliassen-Palm fluxes**. Much insight into extratropical atmospheric circulation can be gained from the Transformed Eulerian Mean (TEM) form of the zonal mean quasi-geostrophic (QG) momentum equation[68,81]:

$$\frac{\partial \bar{u}}{\partial t} = f\bar{v}^* + \frac{1}{a\cos\phi}\nabla\cdot\boldsymbol{F} + \bar{e} \qquad (2)$$

where the overbar represents the zonal mean and the * denotes the TEM circulation:

$$\bar{v}^* = \bar{v} - \frac{\partial}{\partial p}\left[\frac{\overline{v'\theta'}}{\partial\bar{\theta}/\partial p}\right] \qquad (3)$$

$$\bar{w}^* = \bar{w} + \frac{1}{a\cos\phi}\frac{\partial}{\partial\phi}\left[\frac{\overline{v'\theta'}\cos\phi}{\partial\bar{\theta}/\partial p}\right] \qquad (4)$$

where the prime represents departures from the zonal mean, $u$, $v$, $w$ are the zonal, meridional and vertical velocities, $\theta$ is potential temperature, $f$ is the Coriolis parameter, $t$ is time, $\bar{e}$ represents the effects of friction and parameterised processes, and the Eliassen–Palm (EP) flux divergence is given by

$$\nabla\cdot\boldsymbol{F} = \nabla_\phi F_\phi + \nabla_p F_p = \frac{1}{a\cos\phi}\frac{\partial F_\phi\cos\phi}{\partial\phi} + \frac{\partial F_p}{\partial p} \qquad (5)$$

where the QG northward and vertical EP fluxes are given by

$$\{F_\phi, F_p\} = a\cos\phi\left\{-\overline{u'v'}, \frac{\overline{v'\theta'}}{\partial\bar{\theta}/\partial p}f\right\} \qquad (6)$$

Note, that we use the full primitive equation EP fluxes[81] in our analysis but the main effects are captured by the QG form given above.

**Standardisation**. To aid visualisation the TEM circulation and EP fluxes are scaled by dividing by the internal variability of the present-day simulations, computed as the variance across ensemble members, averaged over all of the models, and then square-rooted to obtain the standard deviation. This standardises the magnitude of signals across the depth of the atmosphere and indicates the general sense of the propagation (i.e. north or south, upwards or downwards) but does not necessarily indicate the precise geometric direction.

**Refractive index**. Insight into the propagation of wave activity can be gained by examining the refractive index[106,107]:

$$n_k^2 = \frac{\bar{q}_\phi}{\bar{u}} - \left(\frac{k}{a\cos\phi}\right)^2 - \left(\frac{f}{2NH}\right)^2 \qquad (7)$$

where

$$\bar{q}_\phi = \frac{2\Omega}{a}\cos\phi - \frac{1}{a^2}\left[\frac{(\bar{u}\cos\phi)_\phi}{\cos\phi}\right]_\phi - \frac{f^2}{\rho_0}\left(\rho_0\frac{\bar{u}_z}{N^2}\right)_z \qquad (8)$$

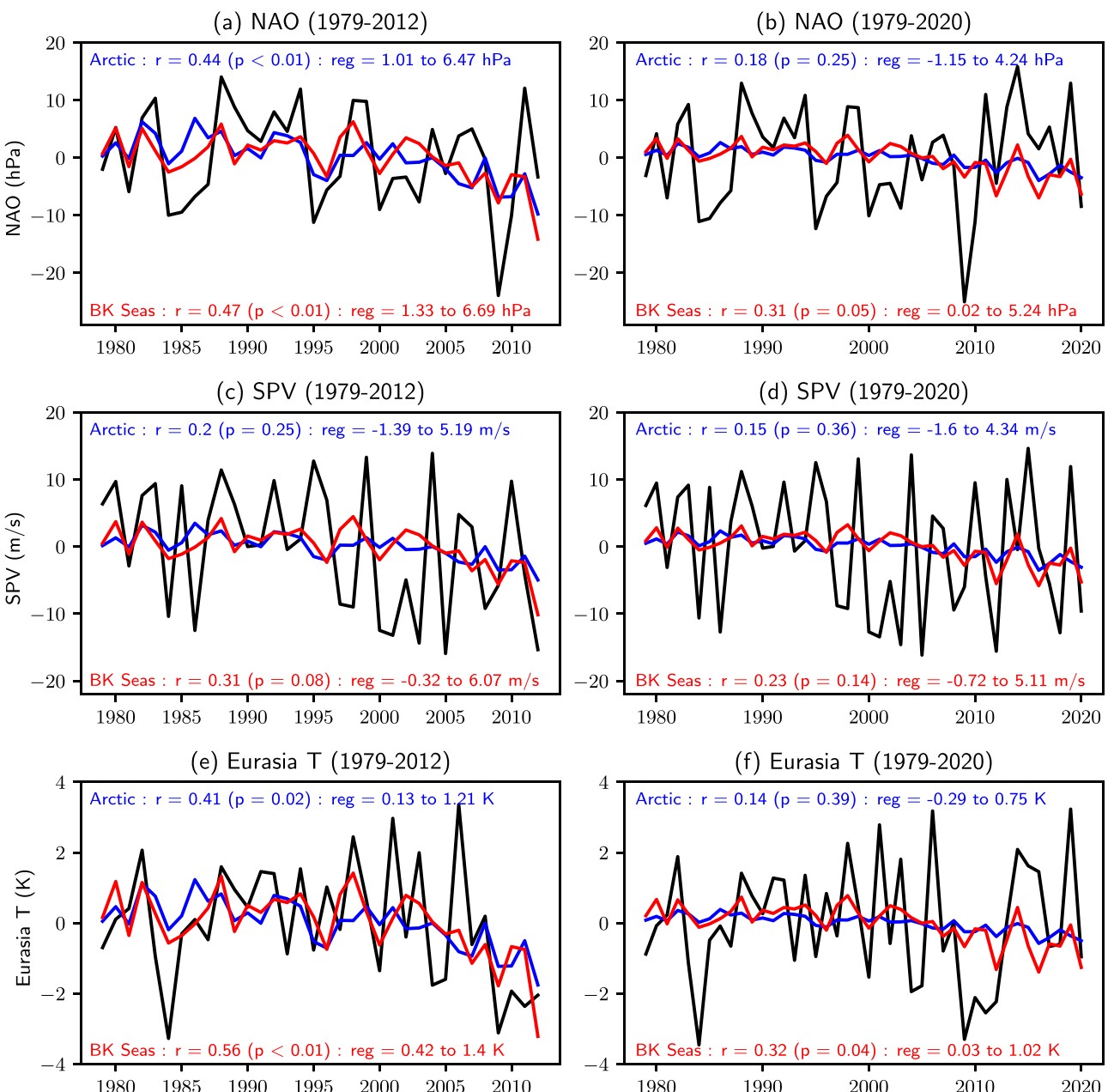

**Fig. 9 Weakened observed relationships. a** Observed winter (DJF) NAO anomaly time-series for the period 1979 to 2012 (black) along with the variability that is linearly related to autumn (September-November) sea ice extent in the Arctic (blue) and Barents-Kara (BK) Seas (red). Pearson correlation (r) and regression coefficients (reg, 95% confidence interval, per standard deviation of sea ice extent) are indicated. **b** As **a** but for the period 1979 to 2020. **c**, **d** As **a**, **b** but for SPV. **e**, **f** As **a**, **b** but for Eurasia T. Indices are defined in Methods.

is the meridional gradient of the zonal mean potential vorticity, and $k$, $N$, $H$ and $\Omega$ denote the zonal wave number, buoyancy frequency, scale height, and Earth rotation frequency respectively. Note, that waves are refracted towards high values of $n_k^2$.

**Eddy feedback**. Eddies are generated by baroclinic processes in the storm tracks and eddy activity propagates horizontally and vertically. Eddies flux angular momentum in the opposite direction to their propagation, driving (accelerating) the zonal wind in their source regions and dragging (decelerating) the zonal wind in their dissipation regions. These interactions with the mean flow can promote or reduce further eddy generation, producing a positive or negative eddy feedback[77,78]. In simple models, it is possible to switch off the effects of eddies allowing eddy feedback to be diagnosed by comparing to the full response[108]. Previous studies have assessed eddy feedback in comprehensive models using lagged relationships in daily data focussing on large scale patterns of variability[109,110]. However, lagged relationships potentially include persistence that is unrelated to feedbacks and may miss fast feedbacks that

occur before the diagnosed lag. They also require analysing large volumes of daily data.

Here, we propose a new approach that can be used with seasonal mean data and does not require lags to be specified. We reason that the fraction of seasonal mean zonal wind variability that is related to eddies will increase as the eddy feedback becomes more positive, and measure the eddy feedback strength as the local correlation squared (i.e. the variance explained) between DJF $\bar{u}$ and $\nabla_\phi F_\phi$ averaged over the mid to upper northern troposphere (25–72°N, 600–200 hPa, the box shown in Fig. 6a). Although this measure is imperfect since it includes eddy driving, we show that it explains some of the differences in modelled responses. For the model simulations, the eddy feedback parameter is calculated across the ensemble members for the present-day simulations, whereas for the reanalyses it is calculated from time series covering the period 1979–2016. The reanalyses eddy feedback parameters are insensitive to removing a linear trend, but we find some sensitivity to the different ways of calculating eddy feedback using models for which time series data are also available (see main text). However, our main conclusions remain valid.

**Constrained estimates**. We use emergent constraints[82,111,112] to estimate the real world response. We seek an observable quantity ($x$) that provides a physical explanation for differences in the model estimates of the response to Arctic sea ice loss ($y_i$, the ensemble mean estimate of the response for model $i$) such that

$$y_i = \bar{y}_i + \beta x_i + \varepsilon_i \tag{9}$$

where the overbar denotes the average of the model estimates, $\beta$ is the slope of the linear regression between $x_i$ and $y_i$, and $\varepsilon_i$ is an identically independently distributed random variable with zero expectation (i.e. noise).

If such a quantity exists, then the simple multi-model ensemble mean (EM i.e. with $\beta = 0$) is an inappropriate estimate of the true response because $\varepsilon_i$ would not be independent of $x$[112]. Instead, the observable response ($y_O$) may be estimated using ensemble regression[112] (ER):

$$y_O = \bar{y}_i + \beta(x_O - \bar{x}_i) \tag{10}$$

where $x_O$ is the observed estimate of $x$. The error variance of the multi-model ensemble mean is

$$S_y^2 = \sigma_\varepsilon^2 \left( \frac{1}{n} + \frac{(x_O - \bar{x}_i)^2}{\sum_{i=1}^{n}(x_i - \bar{x}_i)^2} \right) \tag{11}$$

where $n$ is the number of models and $\sigma_\varepsilon^2$ is the variance of $\varepsilon_i$. The second term in the parentheses accounts for estimation error in regression slope and grows quadratically with the error in the model mean estimate of $x$.

## Data availability

PAMIP datasets analysed during the current study are available from the CMIP data archive https://esgf-node.llnl.gov/projects/cmip6/. Data for HadGEM3-LL are available from https://zenodo.org/record/5127891, and data for OpenIFS are available from https://cera-www.dkrz.de/WDCC/ui/cerasearch datasets DKRZ_LTA_995_ds00003 to DKRZ_LTA_995_ds00008 inclusive. Reanalyses EP fluxes are available from the Centre for Environmental Data Analysis https://catalogue.ceda.ac.uk/uuid/dafbd838e4cc4c68a5ccdd90690ea57f. HadISST1.1 sea ice observations are available from https://www.metoffice.gov.uk/hadobs/hadisst/. ERA5 reanalysis data are available from https://www.ecmwf.int/en/forecasts/datasets/reanalysis-datasets/era5.

## Code availability

The code used during the current study is available from the corresponding author on reasonable request.

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

## Acknowledgements

D.M.S., R.E., L.H., L.S.G., T.J., T.S., X.L., and P.O. were supported by the EU H2020 APPLICATE project (GA727862). The Met Office contribution was also supported by the Met Office Hadley Centre Climate Programme funded by BEIS and Defra and by the UK-China Research and Innovation Partnership Fund through the Met Office Climate Science for Service Partnership (CSSP) China as part of the Newton Fund. J.A.S was supported by NERC grants NE/P006760/1, NE/R005125/1 and NE/V005855/1. G.M and Y.P. were supported by the US Department of Energy, grant number DE-SC0019407. L.S.G was also supported by the Research council of Norway INES project (270061), and the Norwegian e-infrastructure for Research and Education (UNINETT Sigma2) through projects NN2345K, NS2345K and NS9034K. E.M. and D.M. acknowledge the support of the German Federal Ministry of Education and Research through the JPI Climate/JPI Oceans NextG-Climate Science-ROADMAP (FKZ: 01LP2002A) project and of the European Union's Horizon 2020 Programme through the Blue-Action Project (GA727852); and the use of resources from the DKRZ bm0966 and bm1190 projects. C. Deser acknowledges support from the National Center for Atmospheric Research, which is a major facility sponsored by the US National Science Foundation under cooperative agreement 1852977. M.M. was supported by MEXT through the Integrated Research Program for Advancing Climate Models (JPMXD0717935457) and ArCS II (JPMXD1420318865) programs, and by the Environment Research and Technology Development Fund (JPMEERF20192004). J.G.-S. and P.O. were supported by the Spanish Ramón y Cajal' programme (RYC-2016-21181, RYC-2016-22772). B.H. was jointly funded by the Strategic Priority Research Program of the Chinese Academy of Sciences (Grant No. XDA19070404) and the National Natural Science Foundation of China (Grant Nos. 42030602, 91837101). G.G. was supported by the EU H2020 Blue–Action (GA727852) project and uses the HPC resources of TGCC under the allocations 2018-R0040110492 and 2019-A0060107732 made by GENCI. J.S. acknowledges the project L4 of the Collaborative Research Centre TRR 181 Energy Transfers in Atmosphere and Ocean funded by the Deutsche Forschungsgemeinschaft (DFG, German Research Foundation) under Project 274762653.

## Author contributions

D.M.S. led the analysis and writing, R.E. analysed the data. M.B.A., H.A., A.C., S.C., C.D., N.J.D., J.G.S., G.G., L.S.G., S.C.H., B.H., L.H., T.J., J.K., X.L., G.M., E.M., D.M., M.M., R.M., P.O., Y.P., A.A.S., J.A.S., T.S., M.Sig, J.S., L.S., A.W., D.M.S., and R.E. contributed to producing the model results. M.Sea contributed to producing the observed results. All authors commented on the manuscript.

## Competing interests

The authors declare no competing interests.
