## [Peer Review File · Nature Communications]

Robust but weak winter atmospheric circulation response to future Arctic sea ice lossReviewers' Comments:

Reviewer #1:

Remarks to the Author:

In this paper, Smith et al examine the winter atmospheric circulation response to Arctic sea ice loss using results from 16 models contributing to the Polar Amplification Model Intercomparison Project. The authors convincingly show that the ensemble-mean response to winter sea-ice loss under the global-mean warming of 2C is an equatorward shift of the mid-latitude jet in the Northern Hemisphere, associated with cooling over mid-latitudes. The authors also develop an emergent constraint for the atmospheric response to sea-ice loss based on an "eddy feedback". I am very skeptical of the logic behind this proposed emergent constraint, as well as of its robustness, so it is difficult for me to recommend publication of the manuscript at this stage. My concerns about the feedback are listed below, as well as some minor comments.

1. The authors use a 3 stage mechanism to explain the weakening of the upward EP flux, and to explain the "positive eddy feedback" which underlies their emergent constraint. However, this mechanism seems to confuse many aspects of the wave-mean flow problem of jet shifts. For example, in stage 1 the authors state: "Reduced wind shear reduces baroclinic eddy formation, weakening the storm track and reducing F_p at the surface..." But wind shear and baroclinic eddy formation are *both* determined by meridional temperature gradients: near-surface temperature gradients are responsible for the reversal of the PV gradient that drives baroclinic eddy formation. The QG dynamics which the authors appeal to here represent the 2nd-order dynamics on top of the 1st-order geostrophically-balanced flow (i.e., thermal wind). Similarly, the second step of the mechanism describes a circulation that is consistent with the jet shift, but cannot be used to establish causality. I also note that the authors further undermine their conceptual picture with some notation issues, e.g. taking the divergence of scalar quantities (see minor comments below).

I suggest either cutting this section, or scaling the discussion back and simply noting that the near-surface temperature response drives an equatorward jet shift by both shifting the temperature gradient and shifting the baroclinic regional equatorwards. Filling in the details can be left to future studies.

Having said all that, changes in eddy momentum fluxes do reinforce ("feedback" on) jet shifts, but this is true of any jet shift. So, if I've understood correctly, the authors' emergent constraint claims that this set of models will underestimate jet shifts in response to any forcing, not just sea ice loss.

2. Moving on to the emergent constraint itself, I am still not sure how exactly it is calculated. It seems as though the authors correlate the EMF divergence in the box shown in Figure 6a with the zonal-mean zonal wind in this box, so that the metric M is the r^2 value of the correlations for DJF. The correlations use the highest frequency data available, without any averaging. Is this correct? Are the reanalysis data de-trended? It seems like the PAMIP experiments are time-slices without repeating climatology (no trends). Could this affect the comparison with the reanalysis data (which may have some underlying trends)? I am also concerned that the relationships shown in Figure 7 mostly come from one "bad" model: E3SMv1. As discussed e.g., by Brient and Schneider (2016), "bad" outlier models can exert a strong leverage on emergent constraints, yet we should place less weight on them, since they are presumably less realistic. I suggest the authors either use a methodology which damps the impact of these models (like the Brient and Schneider approach) or investigate how their results change when this model is not included in the analysis.

Reference:

Brient, F. and T. Schneider, 2016: Constraints on climate sensitivity from space-based measurements of low-cloud reflection. *Journal of Climate*, 29, 5821-5835.

3. The authors seem to undermine their own emergent constraint at L264-6: "For example, ZWRI is

correlated with the background SPV ($r=0.50$, $p=0.03$, not shown) but in this case the observations are near the middle of the model range so that ER is close to the simple ensemble mean". So one constraint suggests the observations are outside of the model range, and the other puts them in the middle of the model range -- which should we trust? Or should have low confidence in both constraints?

Minor comments:

1. In equation 4, and the discussion of the EP fluxes, there seems to be some confusion regarding vector and scalar quantities. In the middle of equation 5, the authors are taking gradients of scalar variables, not the divergence of vectors (∇F_ϕ not $\nabla \cdot F_\phi$). Also I suggest just writing out the $\frac{\partial F_p}{\partial p}$, otherwise it's confusing whether the gradient operator refers to a horizontal gradient, a vertical gradient or a 3D gradient.
2. At L580 it says "cite" where I'm guessing a citation is meant to go. Also at L644.
3. Not sure what is meant by the sentence at L642-3. In the ensemble mean, $\bar{y} = \beta \bar{x}$.
4. Figure 2: Suggesting using a single colorbar, along the bottom or right side of the figure, to save space.
5. Figure 3c: I'm confused, the contours show F_p , and the arrows show (F_ϕ, F_p) . Is this correct? I also don't understand how the normalization was done from the caption. Could you either write it out or use equations?
6. Figure 4d: what do the solid/dashed gray contours represent?
7. Figure 9: should the caption say the BK curves are red, not green?

Reviewer #2:

Remarks to the Author:

Accept subject to minor revisions

The manuscript is thorough showing the atmospheric dynamic complexity due to loss of sea ice. I was pleased with the discussion of real world physics in addition to just models.

Line 294 I do not understand sentence: since models are able to predict the real world better than themselves

Line 317 I Disagree: "are thus unlikely to drive large impacts in individual winters." It is still possible to have short impact events of one to several weeks in any given year. A seasonal average may still be small

Reviewer #3:

Remarks to the Author:

Using 16 different atmospheric models with more than 3000 ensemble members, this study investigates the transient response of northern hemisphere winter westerlies to future Arctic sea ice loss. Consistent with previous modeling studies, this study finds that the Arctic sea ice loss causes a robust equatorward shift of mid-latitude westerlies: a significant weakening around 50–70N and a slight strengthening around 30–40N. A key finding is that the inter-model differences in zonal-mean

wind responses (ZWRI) can be explained by eddy feedback parameter and the eddy feedback parameter of reanalysis data is about 1.2~3 times larger than in climate models.

I believe a thorough and comprehensive analysis of multi-model simulations can warrant publication at Nature Communications. In particular, this study 1) comprehensively quantifies the sensitivity to sea ice loss by utilizing 16 models with more than 3000 ensembles, 2) provides insight into the sensitivity of zonal-mean wind response to sea ice loss by introducing a zonal wind response index (ZWRI) that can explain the meridional circulation anomalies, and 3) is partly successful in providing an emergent constraint by calculating eddy feedback parameter both for climate models and for reanalysis data.

Specific comments:

It took me considerable time, effort and patience to read through this paper. This is not only because TEM dynamics are difficult to understand but also because this paper tries to deliver too much information.

1) There are two key messages and these two are not closely related to each other. To me, quantifying the multi-model ZWRI by eddy feedback parameter is a key message of this study. However, the abstract emphasizes that the modelled response to Arctic sea ice loss is weak and the relationships between Arctic sea ice and atmospheric circulation have weakened recently.

2) I suggest deleting Figure 9, which is not closely related to previous figures. I understand that the authors want to deliver as much information as possible to educate readers, but please reconsider.

3) Abstract: "the North Atlantic Oscillation response is similar in magnitude and offsets the projected response to increased greenhouse gases, but would only account for around 10% of variations in individual years"

Is this really necessary to include this sentence in the abstract? A previous modelling study pointed out that the equatorward shift of NH westerlies driven by future Arctic sea ice loss is opposed by the response to low-latitude surface warming (see Figure 5 of Blackport and Kushner 2017). They also noted in the abstract that "internal variability can easily contaminate the estimates..."

Small/large, strong/weak are subjective words and the time mean response of westerlies to future Arctic sea ice loss is not necessarily small compared to the westerly response to the future tropical SST warming.

Blackport, R., and P. J. Kushner, 2017: Isolating the Atmospheric Circulation Response to Arctic Sea Ice Loss in the Coupled Climate System. *J. Climate*, 30, 2163–2185.

4) I suggest deleting Figure 4 or move this figure to Supplementary information. I really cannot understand why the October TEM circulation and EP flux anomalies are special and can be interpreted as physical mechanisms. It is well known that summer sea ice loss and the associated increase in Arctic ocean heat content are accompanied by seasonally persistent surface warming. I guess the authors are careful about interpreting the winter surface warming because the winter Arctic warming in observation is not only driven by summer sea ice loss but also by winter circulation anomalies? I think the authors do not need to worry about this issue because this PAMIP experiment is designed to isolate the impact of Arctic sea ice loss from other factors.

5) Lines 629–631: Please explain the difference between eddy driving and eddy feedback.

6) Lines 642–643: I am not sure whether this statement is correct or not. Please consult with a statistician.

7) Line 644: "regressioncite" seems to be a typo.

8) Captions in Figures 5 and 6: Which season? Are they about DJF average?

9) Line 574: "assess the effect of coupling": Does coupling imply ocean coupling?

10) Please write down the definitions of \bar{U} and \bar{T} shown in Figures 3, 4, 5, 6 more in detail. It seems that \bar{T} is zonal-mean temperature anomalies and \bar{U} is zonal-mean zonal wind. How about changing \bar{T} and \bar{U} to $[T]$ and $[U]$?

Many thanks for your comments and suggestions. Please see our replies in blue below.

Reviewer #1 (Remarks to the Author):

In this paper, Smith et al examine the winter atmospheric circulation response to Arctic sea ice loss using results from 16 models contributing to the Polar Amplification Model Intercomparison Project. The authors convincingly show that the ensemble-mean response to winter sea-ice loss under the global-mean warming of 2C is an equatorward shift of the mid-latitude jet in the Northern Hemisphere, associated with cooling over mid-latitudes. The authors also develop an emergent constraint for the atmospheric response to sea-ice loss based on an "eddy feedback". I am very skeptical of the logic behind this proposed emergent constraint, as well as of its robustness, so it is difficult for me to recommend publication of the manuscript at this stage. My concerns about the feedback are listed below, as well as some minor comments.

1. The authors use a 3 stage mechanism to explain the weakening of the upward EP flux, and to explain the "positive eddy feedback" which underlies their emergent constraint. However, this mechanism seems to confuse many aspects of the wave-mean flow problem of jet shifts. For example, in stage 1 the authors state: "Reduced wind shear reduces baroclinic eddy formation, weakening the storm track and reducing F_p at the surface..." But wind shear and baroclinic eddy formation are *both* determined by meridional temperature gradients: near-surface temperature gradients are responsible for the reversal of the PV gradient that drives baroclinic eddy formation.

*We agree, which is why we were careful to include both a reduction in wind shear **and** a reduction in baroclinic eddy formation and F_p in stage 1. To make this clearer we have renamed stage 1 to be "Reduced zonal wind shear and eddy formation".*

The QG dynamics which the authors appeal to here represent the 2nd-order dynamics on top of the 1st-order geostrophically-balanced flow (i.e., thermal wind).

*Again, we agree, which is why stage 1 is predominantly governed by the thermal wind relation whereas stage 2 requires a consideration of the 2nd-order dynamics. Of course, both thermal wind and the QG balance must be satisfied together, and the meridional circulation is the adjustment needed to satisfy thermal wind **and** the changes in eddy fluxes, which both occur in stage 1 as mentioned above. We have amended the text to make it clearer that the resulting flow maintains the thermal wind balance and is consistent with changes in eddy activity.*

Similarly, the second step of the mechanism describes a circulation that is consistent with the jet shift, but cannot be used to establish causality.

We agree that causality cannot be established unequivocally. We now state this and have amended the text to avoid claiming causality. However, the processes that we highlight are all consequences of an imposed reduction in surface meridional temperature gradient and highlight a key role for eddies. Furthermore, the monthly evolution provides some information about how the different processes develop. To make this clearer we have expanded Fig 4 to show November in addition to October.

I also note that the authors further undermine their conceptual picture with some notation issues, e.g. taking the divergence of scalar quantities (see minor comments below).

Apologies, this has now been corrected (see minor comments below) – thanks for spotting this

I suggest either cutting this section, or scaling the discussion back and simply noting that the near-

surface temperature response drives an equatorward jet shift by both shifting the temperature gradient and shifting the baroclinic regional equatorwards. Filling in the details can be left to future studies.

We believe that a detailed understanding of the physical processes is essential for developing a credible emergent constraint. We are therefore loathed to cut this analysis which shows an important role for eddies and hence underpins our proposed constraint. We are conscious however that some of the processes are not particularly easy to understand and we have amended the text to clarify our arguments. We believe this section is now much clearer – but suggestions for further improvements would of course be welcome.

Having said all that, changes in eddy momentum fluxes do reinforce ("feedback" on) jet shifts, but this is true of any jet shift. So, if I've understood correctly, the authors' emergent constraint claims that this set of models will underestimate jet shifts in response to any forcing, not just sea ice loss.

Absolutely. We already link to the "signal-to-noise paradox" which shows that models underestimate the magnitude of predicted atmospheric circulation changes, especially in the north Atlantic, and we hope that our new results will motivate further studies. We have also added a comment that our emergent constraint could apply to other forcings to motivate further work to understand the model spread in future climate projections.

2. Moving on to the emergent constraint itself, I am still not sure how exactly it is calculated. It seems as though the authors correlate the EMF divergence in the box shown in Figure 6a with the zonal-mean zonal wind in this box, so that the metric M is the r^2 value of the correlations for DJF.

Yes, that is correct – though we have now further clarified that M is the local r^2 between zonal mean wind and EMF divergence averaged over the box in Fig 6a to avoid any confusion that we may have averaged the zonal winds and EMF divergences over the box before computing the correlations.

The correlations use the highest frequency data available, without any averaging. Is this correct?

No - we have clarified that the regressions are based on seasonal mean data. This avoids the need to analyse large volumes of high frequency data and allows models for which high frequency data are not available to be included (daily velocities are not top priority in the PAMIP data request). We find strong correlations using seasonal mean data (Fig. 6a) along with a large spread across the models (Fig. 6b) and statistically significant differences in our eddy feedback parameter (Fig. 7). Furthermore, our measure of eddy feedback explains some of the model spread in ZWRI, consistent with the important role of eddies in the physical mechanism. Hence, we argue that eddy feedback may be assessed with seasonal mean data and hope that our simple measure will facilitate future studies of its role in other contexts.

Are the reanalysis data de-trended? It seems like the PAMIP experiments are time-slices without repeating climatology (no trends). Could this affect the comparison with the reanalysis data (which may have some underlying trends)?

This is a good question and in fact we had already checked that detrending makes virtually no difference to the eddy feedback estimation for the reanalyses. We have now included a statement to make this clear – thanks for highlighting this.

I am also concerned that the relationships shown in Figure 7 mostly come from one "bad" model: E3SMv1. As discussed e.g., by Briant and Schneider (2016), "bad" outlier models can exert a strong

leverage on emergent constraints, yet we should place less weight on them, since they are presumably less realistic. I suggest the authors either use a methodology which damps the impact of these models (like the Brient and Schneider approach) or investigate how their results change when this model is not included in the analysis.

Reference:

Brient, F. and T. Schneider, 2016: Constraints on climate sensitivity from space-based measurements of low-cloud reflection. *Journal of Climate*, 29, 5821-5835.

We tested this by removing each model in turn and repeating the regression, as suggested. This is most sensitive to removing E3SMv1 and CanESM5, increasing the p values to 0.16 and 0.07 respectively for ZWRI, and to 0.06 and 0.10 respectively for SPV. Thus, the ZWRI or SPV remains significant at $p \leq 0.07$ when outlying models are removed. We have included these results (and the reference) in the manuscript.

3. The authors seem to undermine their own emergent constraint at L264-6: "For example, ZWRI is correlated with the background SPV ($r=0.50$, $p=0.03$, not shown) but in this case the observations are near the middle of the model range so that ER is close to the simple ensemble mean". So one constraint suggests the observations are outside of the model range, and the other puts them in the middle of the model range -- which should we trust? Or should have low confidence in both constraints?

Apologies for not being clear here. If we had only looked at the relationship between ZWRI and the polar vortex strength we would have concluded that the real-world response was near the middle of the models. But this would be incorrect, given the existence of the constraint based on eddy feedback that is much more strongly related to the physical processes and places the real world towards the upper end of the models. We have now amended the text to make this clearer.

Minor comments:

1. In equation 4, and the discussion of the EP fluxes, there seems to be some confusion regarding vector and scalar quantities. In the middle of equation 5, the authors are taking gradients of scalar variables, not the divergence of vectors (∇F_ϕ not $\nabla \cdot F_\phi$). Also I suggest just writing out the $\frac{\partial F_p}{\partial p}$, otherwise it's confusing whether the gradient operator refers to a horizontal gradient, a vertical gradient or a 3D gradient.

Apologies, this has now been corrected – thanks for spotting this. We decided to use $\nabla_\phi F_\phi$ etc rather than write out $\frac{1}{a \cos \phi} \frac{\partial F_\phi \cos \phi}{\partial \phi}$

2. At L580 it says "cite" where I'm guessing a citation is meant to go. Also at L644.

Corrected – thanks for spotting these

3. Not sure what is meant by the sentence at L642-3. In the ensemble mean, $\bar{y} = \beta \bar{x}$.

We have clarified that the simple multi-model ensemble mean is inappropriate if the noise is not independent of x , as stated by Bracegirdle and Stephenson (2012)

4. Figure 2: Suggesting using a single colorbar, along the bottom or right side of the figure, to save space.

Agreed, figure has been replaced – thanks for this suggestion

5. Figure 3c: I'm confused, the contours show F_p , and the arrows show (F_ϕ, F_p) . Is this correct? I also don't understand how the normalization was done from the caption. Could you either write it out or use equations?

Yes, that is correct – we have adjusted the caption to make it clearer. We have also clarified the standardisation by adding a description in Methods.

6. Figure 4d: what do the solid/dashed gray contours represent?

The contours are unnecessary and have now been removed – thanks for spotting this

7. Figure 9: should the caption say the BK curves are red, not green?

Yes, thanks for spotting this – now corrected

Many thanks for your comments and suggestions. Please see our replies in blue below.

Reviewer #2 (Remarks to the Author):

Accept subject to minor revisions

The manuscript is thorough showing the atmospheric dynamic complexity due to loss of sea ice. I was pleased with the discussion of real world physics in addition to just models.

Line 294 I do not understand sentence: since models are able to predict the real world better than themselves

We have clarified this: "This has been referred to as the "signal-to-noise paradox" (Scaife and Smith 2018) since models are unexpectedly able to predict the real world better than they can predict one of their own ensemble members".

Line 317 I Disagree: "are thus unlikely to drive large impacts in individual winters." It is still possible to have short impact events of one to several weeks in any given year. A seasonal average may still be small

Thanks – we have clarified that large seasonal mean impacts are unlikely

Many thanks for your comments and suggestions. Please see our replies in blue below.

Reviewer #3 (Remarks to the Author):

Using 16 different atmospheric models with more than 3000 ensemble members, this study investigates the transient response of northern hemisphere winter westerlies to future Arctic sea ice loss. Consistent with previous modeling studies, this study finds that the Arctic sea ice loss causes a robust equatorward shift of mid-latitude westerlies: a significant weakening around 50–70N and a slight strengthening around 30–40N. A key finding is that the inter-model differences in zonal-mean wind responses (ZWRI) can be explained by eddy feedback parameter and the eddy feedback parameter of reanalysis data is about 1.2~3 times larger than in climate models.

I believe a thorough and comprehensive analysis of multi-model simulations can warrant publication at Nature Communications. In particular, this study 1) comprehensively quantifies the sensitivity to sea ice loss by utilizing 16 models with more than 3000 ensembles, 2) provides insight into the sensitivity of zonal-mean wind response to sea ice loss by introducing a zonal wind response index (ZWRI) that can explain the meridional circulation anomalies, and 3) is partly successful in providing an emergent constraint by calculating eddy feedback parameter both for climate models and for reanalysis data.

Specific comments:

It took me considerable time, effort and patience to read through this paper. This is not only because TEM dynamics are difficult to understand but also because this paper tries to deliver too much information.

Many thanks for your patience and perseverance. We accept that the paper contains a lot of information, but we believe that a detailed description of the physical processes is needed to justify the constraint, and significantly adds to previous studies. Following your comments, we have clarified some of the text, broken down some of the paragraphs into more digestible pieces, and added text to clarify our arguments. We hope these improvements make the paper much easier to read.

1) There are two key messages and these two are not closely related to each other. To me, quantifying the multi-model ZWRI by eddy feedback parameter is a key message of this study. However, the abstract emphasizes that the modelled response to Arctic sea ice loss is weak and the relationships between Arctic sea ice and atmospheric circulation have weakened recently.

We believe that both messages are important for addressing the perceived disagreement between observations and models (highlighted recently in ref 5): we show both that there is a robust response in models, and that it is consistent with observations when model biases and the latest observational data are taken into account. We also highlight that the response is weak compared to interannual variability, which partly explains why it has been so difficult to diagnose in previous observational and modelling studies.

2) I suggest deleting Figure 9, which is not closely related to previous figures. I understand that the authors want to deliver as much information as possible to educate readers, but please reconsider.

Whether models and observations disagree is a key part of the debate, as highlighted in ref 5. We considered removing Fig 9 as suggested, and simply referring to Blackport and Screen 2020 (ref 7) who also show that observed relationships have weakened recently. However, Blackport and Screen do not provide a quantification of the response that can be compared directly with our model results. Hence, we believe Fig 9 is needed to assess whether models and observations are consistent.

3) Abstract: "the North Atlantic Oscillation response is similar in magnitude and offsets the projected response to increased greenhouse gases, but would only account for around 10% of variations in individual years"

Is this really necessary to include this sentence in the abstract? A previous modelling study pointed out that the equatorward shift of NH westerlies driven by future Arctic sea ice loss is opposed by the response to low-latitude surface warming (see Figure 5 of Blackport and Kushner 2017). They also noted in the abstract that "internal variability can easily contaminate the estimates..."

Small/large, strong/weak are subjective words and the time mean response of westerlies to future Arctic sea ice loss is not necessarily small compared to the westerly response to the future tropical SST warming.

Blackport, R., and P. J. Kushner, 2017: Isolating the Atmospheric Circulation Response to Arctic Sea Ice Loss in the Coupled Climate System. *J. Climate*, 30, 2163–2185.

What is new relative to Blackport and Kushner and other studies is that we have quantified the magnitude of the response with an emergent constraint. This sentence puts that in context, and the most relevant comparisons are with interannual variability and the long-term projected change to increases in greenhouse gases.

4) I suggest deleting Figure 4 or move this figure to Supplementary information. I really cannot understand why the October TEM circulation and EP flux anomalies are special and can be interpreted as physical mechanisms. It is well known that summer sea ice loss and the associated increase in Arctic ocean heat content are accompanied by seasonally persistent surface warming. I guess the authors are careful about interpreting the winter surface warming because the winter Arctic warming in observation is not only driven by summer sea ice loss but also by winter circulation anomalies? I think the authors do not need to worry about this issue because this PAMIP experiment is designed to isolate the impact of Arctic sea ice loss from other factors.

Figure 4 is key for understanding the physical mechanism and hence for developing the emergent constraint. Many studies have pointed to an increase in upward wave flux that reduces the polar vortex, but the reason for this increase has not been understood before, and it is counterintuitive given the expected weakening of the storm tracks which are the source of the waves. It is possible to increase upward wave activity directly by increasing the zonal asymmetries in the sea ice region, and there is some evidence for this (positive values near the surface at latitudes greater than 80N in Fig 3c). However, by far the strongest increase in upward wave flux occurs around 40-50N, and this is also the pathway into the stratosphere which has been highlighted to begin in October in other studies. Figure 4 shows that the response evolves from the expected reduction in upward wave activity in October (consistent with reduced storm tracks) to the DJF equatorward shift, and that this equatorward shift is consistent with an eddy-driven meridional circulation, highlighting the potential role of eddy feedback that is used in the emergent constraint.

Many thanks for your comment. In response we have strengthened our discussion, and hope this has improved the paper.

5) Lines 629–631: Please explain the difference between eddy driving and eddy feedback.

We have clarified the difference.

6) Lines 642–643: I am not sure whether this statement is correct or not. Please consult with a statistician.

This is stated by Bracegirdle and Stephenson (2012) – reference now added

7) Line 644: "regressioncite" seems to be a typo.

Corrected – thanks for spotting this

8) Captions in Figures 5 and 6: Which season? Are they about DJF average?

Yes – now clarified in the captions – thanks for pointing this out

9) Line 574: "assess the effect of coupling": Does coupling imply ocean coupling?

We have clarified that this refers to ocean-atmosphere coupling – thanks for pointing this out

10) Please write down the definitions of \bar{U} and \bar{T} shown in Figures 3, 4, 5, 6 more in detail. It seems that \bar{T} is zonal-mean temperature anomalies and \bar{U} is zonal-mean zonal wind. How about changing \bar{T} and \bar{U} to $[T]$ and $[U]$?

Yes, \bar{u} and \bar{T} are simply the zonal means. We have clarified this (line 95) and prefer to keep this commonly used notation.

Reviewers' Comments:

Reviewer #1:

Remarks to the Author:

This is a resubmission of a study by Smith et al on the winter atmospheric circulation response to future Arctic sea ice loss. In the previous round of review, I was skeptical of the mechanism proposed by Smith et al to explain the equatorward jet shift seen in response to Arctic sea ice loss and of the logic behind their emergent constraint. The authors have done a good job explaining their thinking and clarifying the text. However, I still find that the discussion of the proposed mechanism causes more problems than it's worth, and would suggest simplifying the discussion.

Major Comments:

1. The authors want to propose a dynamical mechanism to justify their use of the "eddy feedback" in the emergent constraint. While this is admirable, as it stands, the discussion raises questions for the reader that aren't answered in the main text. For example, what changes between October and November to kick off the meridional circulation (Reviewer #3 also asked this question in the initial reviews)? What role does surface friction play? And why should the horizontal momentum fluxes respond after the changes in the vertical wave activity? Changes in [u] driven by temperature changes will also affect horizontal wave propagation, why can't these change simultaneously? In fact, the zonal wind changes are quite barotropic (Fig. 4) suggesting that its not just the vertical wave activity that changes.

Given that changes in horizontal wave propagation (i.e., the "eddy feedback" referred to here) are well known to contribute to jet shifts (see e.g., the review paper by Shaw, 2019), it seems safe for the authors to proceed with their emergent constraint without trying to explain exactly what causes the jet to move equatorward.

Reference: Shaw, T. A., 2019: Mechanisms of future predicted changes in the zonal mean mid-latitude circulation, Current Climate Change Reports, 10.1007/s40641-019-00145-8.

2. In the response to the reviewers, the authors state that "...the ZWRI or SPV remains significant at $p \leq 0.07$ when outlying models are removed". But just above that they state that removing E3SMv1 increases the p value to 0.16 for ZWRI, which is clearly > 0.07 . Thus I am still not convinced that the emergent constraint isn't just coming from one bad model (the one furthest from the observations).

Minor comments:

1. L76: I might have missed it, but are the same SI loss and SST warming applied to all the models? Or is each model forced by its own response? In other words, what is meant by "values expected if global temperatures rise by 2C"?

2. L114: Suggest "meridional circulation" -> "meridional overturning circulation", since the air doesn't just move north-south.

3. L168-70: Not sure what is meant by: "However, zonal wind cannot increase near the surface because wind shear must be reduced according to the thermal wind response to the imposed surface temperature gradient." Wouldn't the vertical wind shear be further reduced if the zonal wind near the surface increased? It would also be good to discuss the role of friction in this picture -- the friction (and hence the surface winds) must balance the vertically integrated momentum flux divergence.

Reviewer #2:

Remarks to the Author:

The authors now have the right conclusion at the end:

Line 337 and are thus unlikely to drive large seasonal mean impacts in individual winters.

However I disagree then with their earlier statement that intermittency does not reflect a causal link. There is a causal link during the event, it is just that these are average out over a season
Consistent with recent evidence that the observed relationships are modest and intermittent
Line 329 and may not reflect a causal link with Arctic sea ice98–100 330 .

Also given the seasonal average statement in the conclusion, I disagree with the last line in the Abstract as it is too strong. There is a sea ice connection during events, just not large on a seasonal average

We further find that

line 41 relationships between Arctic sea ice and atmospheric circulation have weakened recently in
line 42 observations and are no longer inconsistent with those in models.

Reviewer #3:

Remarks to the Author:

The manuscript has been improved. This study nicely quantifies the zonal-mean zonal wind responses to the sea ice loss across 16 different models and proposed an emergent constraint by calculating the eddy feedback parameter. This is a very important result that can resolve the long debate on the impact of winter Arctic sea ice loss on the mid-latitude westerlies and climate.

A potential weakness is that the robustness of the eddy feedback parameter is somewhat questionable because the correlation coefficient is not very high (only around 0.5) and becomes statistically less significant when the outlier model(s) is not used. However, I believe this weakness is not critical and this manuscript should be published. If this study is not acceptable, what else can be published at Nature Communications?

Having said that, I guess Figure 4 and the associated discussions are not important. I personally think that October & November EP flux vectors do not provide any clue on the causality of the equatorward shift of westerlies in DJF. As the season progresses from autumn to winter, northern winter stationary waves rapidly strengthen... So, I am a little worried whether October circulation anomalies are analogous to winter circulation dynamics.

Many thanks for your comments and suggestions. Please see our replies in blue below.

Reviewer #1 (Remarks to the Author):

This is a resubmission of a study by Smith et al on the winter atmospheric circulation response to future Arctic sea ice loss. In the previous round of review, I was skeptical of the mechanism proposed by Smith et al to explain the equatorward jet shift seen in response to Arctic sea ice loss and of the logic behind their emergent constraint. The authors have done a good job explaining their thinking and clarifying the text. However, I still find that the discussion of the proposed mechanism causes more problems than it's worth, and would suggest simplifying the discussion.

Thanks for your comments. Part of reason for the long-lasting debate over the mid-latitude impacts of Arctic sea ice is that the physical processes have not been understood in detail. The PAMIP simulations, with more than 3000 ensemble members and 16 different models, provide an unprecedented opportunity to identify the most important processes that are robustly simulated by the majority of models. We summarise an extensive investigation of the month by month evolution and highlight the three processes that are most clearly operating and that can be easily related to the underlying physical principles. We accept that we do not explain every aspect of the circulation response, and we have now made this even clearer. We have also simplified the discussion as far as possible and now refer to "processes" instead of "stages" to avoid any confusion that the processes occur in isolation or in strict order. We appreciate your comments but firmly believe that the robust processes we highlight provide an important step forward and hope that the clarifications we have made are now acceptable.

Major Comments:

1. The authors want to propose a dynamical mechanism to justify their use of the "eddy feedback" in the emergent constraint. While this is admirable, as it stands, the discussion raises questions for the reader that aren't answered in the main text. For example, what changes between October and November to kick off the meridional circulation (Reviewer #3 also asked this question in the initial reviews)?

We highlight the clearest process which is that meridional overturning circulation is initiated by the positive $\nabla_p F_p$ just above the surface (which is an inevitable consequence of reduced F_p at the surface). A positive $\nabla_p F_p$ is clearly seen in October and November during which time the meridional overturning circulation develops, but explaining the timescale for this development is beyond the scope of our study.

What role does surface friction play?

We do not see an active role for changes in friction. Changing the sea ice will of course change the surface drag at high latitudes, but there is no clear evidence that this plays an important role compared to the processes that we highlight.

And why should the horizontal momentum fluxes respond after the changes in the vertical wave activity?

Of course, horizontal momentum fluxes respond as well, but the changes are smaller and $\nabla \cdot F$ is dominated by $\nabla_p F_p$ (as shown in the figure below). We highlight the most important processes but have now made it clearer that other processes are also operating.

Changes in [u] driven by temperature changes will also affect horizontal wave propagation, why can't these change simultaneously?

Horizontal wave propagation does change simultaneously but changes in refractive index are less robust across the models than the processes we highlight.

In fact, the zonal wind changes are quite barotropic (Fig. 4) suggesting that its not just the vertical wave activity that changes.

We agree that we do not explain why the ensemble mean response is barotropic – though this is less clear in individual models in October. We now state more clearly that we do not explain all aspects of the circulation response. Note that the data are available for interested researchers to investigate further.

Given that changes in horizontal wave propagation (i.e., the "eddy feedback" referred to here) are well known to contribute to jet shifts (see e.g., the review paper by Shaw, 2019), it seems safe for the authors to proceed with their emergent constraint without trying to explain exactly what causes the jet to move equatorward.

Reference: Shaw, T. A., 2019: Mechanisms of future predicted changes in the zonal mean mid-latitude circulation, Current Climate Change Reports, 10.1007/s40641-019-00145-8.

Thanks for the useful reference which we now cite. However, we believe that the processes we highlight provide a valuable contribution to the debate on the mid-latitude response to Arctic sea ice.

2. In the response to the reviewers, the authors state that "...the ZWRI or SPV remains significant at $p \leq 0.07$ when outlying models are removed". But just above that they state that removing E3SMv1 increases the p value to 0.16 for ZWRI, which is clearly > 0.07 . Thus I am still not convinced that the emergent constraint isn't just coming from one bad model (the one furthest from the observations).

Removing E3SMv1 increases the p value to 0.16 for ZWRI, but the p value for SPV remains below 0.07 – apologies if this was not clear. We accept that there are uncertainties and we have been completely open about these. We believe the strength of our constraint lies in its close connection to the physical processes. Furthermore, this study developed over many months during which time additional models and ensemble member became available, yet the emergent constraint remained significant, increasing our confidence that it is robust.

Minor comments:

1. L76: I might have missed it, but are the same SI loss and SST warming applied to all the models? Or is each model forced by its own response? In other words, what is meant by "values expected if global temperatures rise by 2C"?

Yes – now clarified.

2. L114: Suggest "meridional circulation" -> "meridional overturning circulation", since the air doesn't just move north-south.

Agreed, now corrected – thanks.

3. L168-70: Not sure what is meant by: "However, zonal wind cannot increase near the surface because wind shear must be reduced according to the thermal wind response to the imposed surface temperature gradient." Wouldn't the vertical wind shear be further reduced if the zonal wind near the surface increased? It would also be good to discuss the role of friction in this picture -- the friction (and hence the surface winds) must balance the vertically integrated momentum flux divergence.

We have now simplified this by saying "zonal wind tends to be reduced in response to the imposed weakening of the surface temperature gradient". We do not see an active role for friction (see reply above) so do not include this in order to keep the discussion as simple as possible.

Many thanks for your comments and suggestions. Please see our replies in blue below.

Reviewer #2 (Remarks to the Author):

The authors now have the right conclusion at the end:

Line 337 and are thus unlikely to drive large seasonal mean impacts in individual winters.

However I disagree then with their earlier statement that intermittency does not reflect a causal link. There is a causal link during the event, it is just that these are average out over a season
Consistent with recent evidence that the observed relationships are modest and intermittent
Line 329 and may not reflect a causal link with Arctic sea ice98-100 330 .

Also given the seasonal average statement in the conclusion, I disagree with the last line in the Abstract as it is too strong. There is a sea ice connection during events, just not large on a seasonal average

We further find that

line 41 relationships between Arctic sea ice and atmospheric circulation have weakened recently in
line 42 observations and are no longer inconsistent with those in models.

The debate we are addressing is how the declining long-term trend in Arctic sea ice is affecting mid-latitude atmospheric circulation. Hence, we assess monthly to seasonal signals rather than individual events, in common with the vast majority of previous studies. If monthly to seasonal signals are weak then the influence of Arctic sea ice trends on the frequency of individual events would also be expected to be weak. We state clearly on line 81 and in the figure captions that our results refer to seasonal or monthly means.

Many thanks for your comments and suggestions. Please see our replies in blue below.

Reviewer #3 (Remarks to the Author):

The manuscript has been improved. This study nicely quantifies the zonal-mean zonal wind responses to the sea ice loss across 16 different models and proposed an emergent constraint by calculating the eddy feedback parameter. This is a very important result that can resolve the long debate on the impact of winter Arctic sea ice loss on the mid-latitude westerlies and climate. A potential weakness is that the robustness of the eddy feedback parameter is somewhat questionable because the correlation coefficient is not very high (only around 0.5) and becomes statistically less significant when the outlier model(s) is not used. However, I believe this weakness is not critical and this manuscript should be published. If this study is not acceptable, what else can be published at Nature Communications?

Having said that, I guess Figure 4 and the associated discussions are not important. I personally think that October & November EP flux vectors do not provide any clue on the causality of the equatorward shift of westerlies in DJF. As the season progresses from autumn to winter, northern winter stationary waves rapidly strengthen... So, I am a little worried whether October circulation anomalies are analogous to winter circulation dynamics.

The PAMIP simulations, with more than 3000 members and 16 different models, provide an unprecedented opportunity to investigate the evolution of the response and highlight the key processes that are robustly simulated by the majority of models. We believe this to be an important part of our study given (1) the need to understand the processes in order to derive an emergent constraint, and (2) the lack of consensus in previous studies over the physical mechanisms through which Arctic sea ice may influence the mid-latitudes.

Reviewers' Comments:

Reviewer #1:

Remarks to the Author:

The authors have addressed my concern and I recommend publication.

Reviewer #1 (Remarks to the Author):

The authors have addressed my concern and I recommend publication.

Many thanks for your time and constructive comments.